# Metazoan evolution of glutamate receptors reveals unreported phylogenetic groups and divergent lineage-specific events

David Ramos-Vicente[1,2], Jie Ji[3], Esther Gratacòs-Batlle[4], Gemma Gou[1,2], Rita Reig-Viader[1,2], Javier Luís[1,2], Demian Burguera[5], Enrique Navas-Perez[5], Jordi García-Fernández[5], Pablo Fuentes-Prior[6], Hector Escriva[7], Nerea Roher[3], David Soto[4], Àlex Bayés[1,2]*

[1]Molecular Physiology of the Synapse Laboratory, Biomedical Research Institute Sant Pau, Barcelona, Spain; [2]Universitat Autònoma de Barcelona, Barcelona, Spain; [3]Institute of Biotechnology and Biomedicine, Department of Cell Biology, Animal Physiology and Immunology, Universitat Autònoma de Barcelona, Barcelona, Spain; [4]Neurophysiology Laboratory, Department of Biomedicine, Medical School, August Pi i Sunyer Biomedical Research Institute, Institute of Neurosciences, Universitat de Barcelona, Barcelona, Spain; [5]Department of Genetics, School of Biology, Institut de Biomedicina, University of Barcelona, Barcelona, Spain; [6]Molecular Bases of Disease, Biomedical Research Institute Sant Pau, Hospital de la Santa Creu i Sant Pau, Barcelona, Spain; [7]Sorbonne Université, CNRS, Biologie Intégrative des Organismes Marins, Banyuls-sur-Mer, France

*For correspondence:
abayesp@santpau.cat

Competing interests: The authors declare that no competing interests exist.

**Abstract** Glutamate receptors are divided in two unrelated families: ionotropic (iGluR), driving synaptic transmission, and metabotropic (mGluR), which modulate synaptic strength. The present classification of GluRs is based on vertebrate proteins and has remained unchanged for over two decades. Here we report an exhaustive phylogenetic study of GluRs in metazoans. Importantly, we demonstrate that GluRs have followed different evolutionary histories in separated animal lineages. Our analysis reveals that the present organization of iGluRs into six classes does not capture the full complexity of their evolution. Instead, we propose an organization into four subfamilies and ten classes, four of which have never been previously described. Furthermore, we report a sister class to mGluR classes I-III, class IV. We show that many unreported proteins are expressed in the nervous system, and that new Epsilon receptors form functional ligand-gated ion channels. We propose an updated classification of glutamate receptors that includes our findings.
DOI: https://doi.org/10.7554/eLife.35774.001

## Introduction

Glutamate is the principal excitatory neurotransmitter in the central nervous system of animals (*Fonnum, 1984*; *Danbolt, 2001*; *Pascual-Anaya and D'Aniello, 2006*). It acts on two families of structurally unrelated receptors: ionotropic glutamate receptors (iGluRs), which are ligand-gated ion channels and G-protein coupled receptors (GPCRs), known as metabotropic glutamate receptors (mGluRs) (*Sobolevsky et al., 2009*; *Conn and Pin, 1997*). While fast excitatory neurotransmission is mediated by iGluRs, metabotropic receptors modulate synaptic transmission strength. iGluRs are formed by four subunits, which can be traced back to bacteria (*Tikhonov and Magazanik, 2009*).

**eLife digest** Nerve cells or neurons communicate with each other by releasing specific molecules in the gap between them, the synapses. The sending neuron passes on messages through packets of chemicals called neurotransmitters, which are picked up by the receiving cell with the help of receptors on its surface. Neurons use different neurotransmitters to send different messages, but one of the most common ones is glutamate.

There are two families of glutamate receptors: ionotropic receptors, which can open or close ion channels in response to neurotransmitters and control the transmission of a signal, and metabotropic receptors, which are linked to a specific protein and control the strength of signal.

Our understanding of these two receptor families comes from animals with backbones, known as vertebrates. But the receptors themselves are ancient. We can trace the first family back as far as bacteria and the second back to single-celled organisms like amoebas. Vertebrates have six classes of ionotropic and three classes of metabotropic glutamate receptor. But other multi-celled animals also have these receptors, so this picture may not be complete.

Here, Ramos-Vicente et al. mapped all major lineages of animals to reveal the evolutionary history of these receptors to find out if the receptor families became more complicated as brain power increased. The results showed that the glutamate receptors found in vertebrates are only a fraction of all the types that exist. In fact, before present-day animal groups emerged, the part of the genome that holds the ionotropic receptor genes duplicated three times. This formed four receptor subfamilies, and our ancestors had all of them. Across the animal kingdom, there are ten, not six, classes of ionotropic receptors and there is an extra class of metabotropic receptors. But only two subfamilies of ionotropic and three out of four metabotropic receptor classes are still present in vertebrates today.

The current classification of glutamate receptors centers around vertebrates, ignoring other animals. But this new data could change that. A better knowledge of these new receptors could aid neuroscientists in better understanding the nervous system. And, using this technique to study other families of proteins could reveal more missing links in evolution.

DOI: https://doi.org/10.7554/eLife.35774.002

The current classification of iGluR subunits includes six classes: α-amino-3-hydroxy-5-methyl-4-isoxazolepropionic acid (AMPA) receptors, Kainate receptors, *N*-methyl-D-aspartate (NMDA) receptors (actually comprising three classes: NMDA1-3) and Delta receptors (*Traynelis et al., 2010*). iGluR subunits of the same class assemble into homo- or heterotetramers (*Karakas and Furukawa, 2014*; *Kumar et al., 2011*) and their ligand selectivity is dictated by a small number of residues located in the ligand-binding domain (*Traynelis et al., 2010*). Accordingly, NMDA subunits GluN1 and GluN3 as well as the Delta subunit GluD2 bind glycine and D-serine, while all subunits from the AMPA and Kainate classes bind glutamate (*Traynelis et al., 2010*; *Kristensen et al., 2016*). Metabotropic glutamate receptors are class C GPCRs and as such are formed by a single polypeptide. mGluRs also appeared before the emergence of metazoans, being present in unicellular organisms such as the amoeba *Dictyostellium discoideum* (*Taniura et al., 2006*). mGluRs are presently organized into three classes (I, II and III) and all their members respond to glutamate (*Conn and Pin, 1997*; *Pin et al., 2003*).

While the phylogeny of the two families of GluRs is well characterized in vertebrates, that of the entire animal kingdom is only poorly understood. The few studies on iGluR evolution outside vertebrates concentrate on a few phyla, leaving many proteins unclassified (*Greer et al., 2017*; *Brockie et al., 2001*; *Janovjak et al., 2011*; *Kenny and Dearden, 2013*). Similarly, the vast majority of mGluRs described so far fall into the three classes described in vertebrates (*Krishnan et al., 2013*; *Kucharski et al., 2007*; *Dillon et al., 2006*). Although, the existence of three insect mGluRs that cluster apart from classes I-III led to propose the existence of a fourth class (*Mitri et al., 2004*). Here we present what to our knowledge is the most comprehensive phylogenetic study of ionotropic and metabotropic GluRs along the animal kingdom. We have favored the use of more slow-evolving species for the construction of phylogenetic trees. These species are particularly amenable to phylogenetics (*Simakov et al., 2013*; *Simakov et al., 2015*; *Putnam et al., 2007*) as they arguably

present lower rates of molecular evolution than other organisms. Our work shows that metazoan evolution of GluRs is much more complex than previously thought. iGluRs present an overall organization into four subfamilies that were already present in the last ancestor of all metazoans. Vertebrate species only retain members of two of these subfamilies. Furthermore, we identify many lineage-specific gains, losses or expansions of GluR phylogenetic groups. Finally, we present experimental evidence showing that unreported GluRs found in the basally divergent chordate *Branchiostoma lanceolatum* (amphioxus) are highly expressed in the nervous system and that members of the unreported Epsilon subfamily, the most phylogenetically spread among unreported groups, can form functional ligand-gated ion channels.

## Results

### Phylogenetics of metazoan ionotropic glutamate receptors reveals four subfamilies, unreported classes and lineage-specific evolutionary dynamics

We have performed a systematic phylogenetic study of iGluR evolution across the animal kingdom. To increase the confidence on iGluRs evolutionary history phylogenetic trees have been generated using two independent methods (Bayesian inference and Maximum-likelihood (ML), *Figure 1* and *Figure 1—figure supplement 1*). Our analysis indicates that the family of iGluRs experienced key duplication events that define its present organization into four previously unreported subfamilies, of which two contain the extensively studied vertebrate classes. Assuming ctenophores as the sister group to all other animals (*Moroz et al., 2014*; *Ryan et al., 2013*), our data suggest that the three major duplication events leading to this four subfamilies occurred before the divergence of current animal phyla (see *Figure 2* for a summary scheme of iGluRs evolution). The first of these duplications produced the separation of the Lambda subfamily, the second lead to divergence of the NMDA subfamily and the third to the split between Epsilon and AKDF subfamilies.

The Lambda subfamily is the most phylogenetically restricted, as we could only identify it in porifers. Thus, Lambda would have been lost in two occasions, in the lineage of ctenophores and in a common ancestor of placozoans, cnidarians and bilaterals. On the other hand, the Epsilon subfamily is the best represented among non-bilaterians, being present in all non-bilaterian phyla investigated. Including in porifers, although we could only identify one Epsilon in sponges, GluE_Ifa from the demosponge *Ircinia fasciculata*. Our data also indicate that this subfamily has been lost in multiple occasions along metazoan evolution, as we could not find it in the protostome, echinoderm or vertebrate species investigated. Interestingly, all ctenophore iGluRs identified, which have been previously reported (*Alberstein et al., 2015*), belong to the Epsilon subfamily. Thus, this phylum would have lost NMDA, Lambda and AKDF proteins. Contrarily, ctenophores would have experienced an important expansion of Epsilon iGluRs, as we report 17 and 10 of these proteins in the two species with genomic information available, *M. leidyi* and *P. bachei*, respectively.

Although we have not identified NMDA receptors in ctenophores, porifers and placozoans our analysis indicates that this subfamily was already present in the last common ancestor of metazoans. This is because the topology of the tree shows that NMDAs appear in the phylogeny at the same level as the Epsilon subfamily, which has representatives in all non-bilateral phyla. According to our data, NMDA1s on the one hand and NMDA2s and NMDA3s on the other contain members of the cnidarian phylum. Although we have only been able to identify one member more closely related to NMDA2 and NMDA3 than NMDA1 (GluN2/3_Nve), its position in the phylogeny is very well supported by both analyses performed. This indicates that a specific duplication occurred in the ancestor of bilaterians originating NMDA2s and NMDA3s. Moreover, we have also identified a cnidarian-specific NMDA class, that we have termed NMDA-Cnidaria, this class presents representative proteins in 3 of the four species investigated. Among bilaterals we have observed conservation of all NMDA classes with the exception of NMDA2s in echinoderms, which are absent from the two species examined. Interestingly, studied cnidarian species substantially expanded their NMDA subfamily repertoire, with at least six members in *Nematostella vectensis*.

In bilaterians the AKDF subfamily diversified into the known AMPA, Kainate and Delta classes, but also into a fourth new class that we have termed Phi. The phylogenetic spread of these classes is quite variable, as AMPA and Kainate are in all bilateral phyla investigated but Delta and Phi are

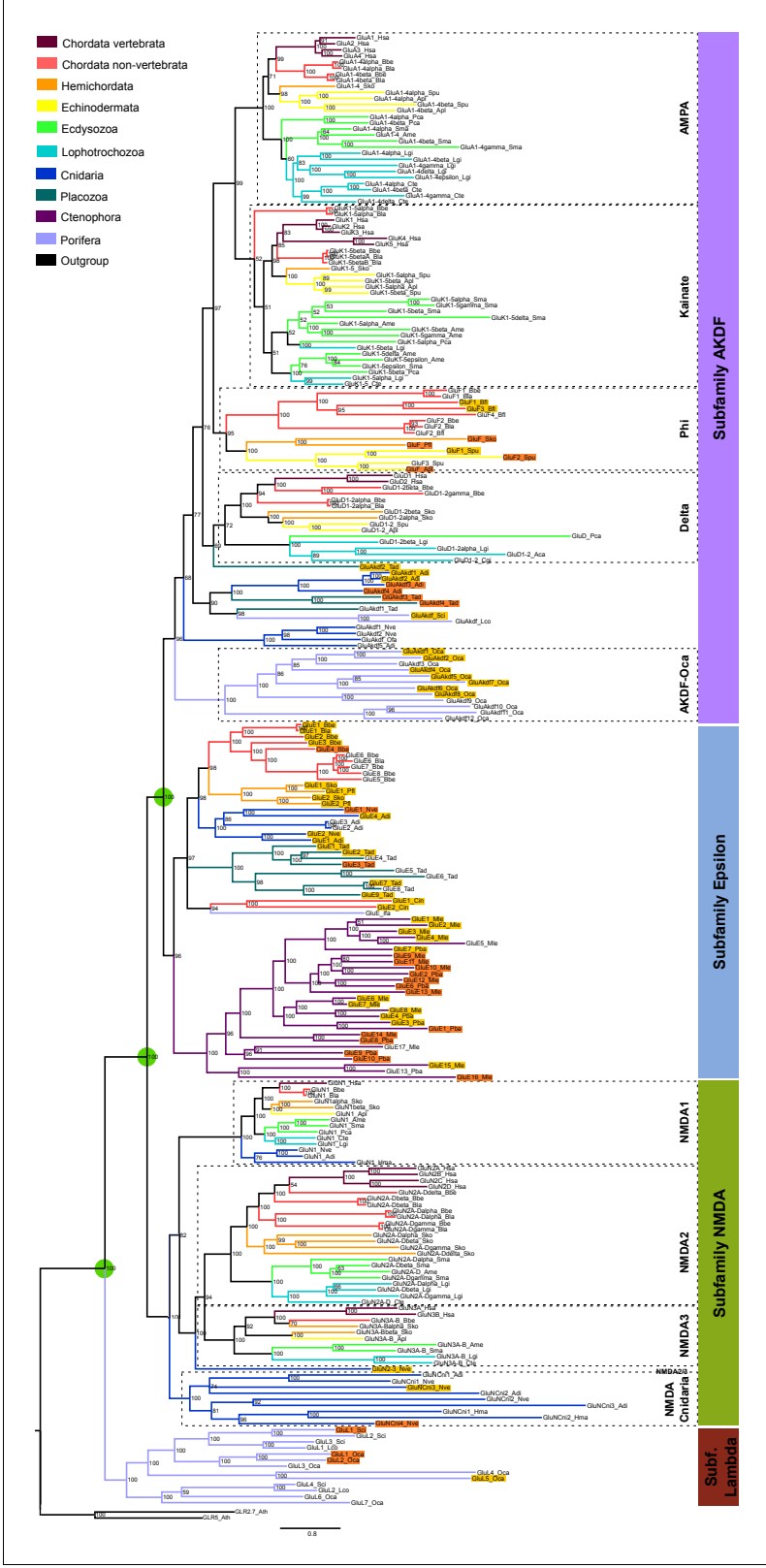

**Figure 1.** Bayesian phylogeny of metazoan ionotropic glutamate receptors. Ionotropic glutamate receptor subfamilies are indicated in colored boxes at the right. Sequences belonging to the same class are highlighted together by dashed lines and the class name is also shown. Green circles highlight the three duplications occurred before the divergence of the ctenophore lineage that lead to these four subfamilies. Posterior probabilities are

*Figure 1 continued*

shown at tree nodes and protein names at the end of each branch. Tree branches are colored based on phylum, as indicated in the legend. For unreported phylogenetic groups, names of proteins predicted to bind glycine or glutamate are highlighted in yellow or orange, respectively. Protein names from non-vertebrate species are composed of four parts: (i) 'GluR#', where # is a code denoting class or subfamily (A, AMPA; K, Kainate; F, Phi; D, Delta; Akdf, AKDF; E, Epsilon; N, NMDA and L, Lambda); (ii) a number, or range of numbers, denoting orthologous vertebrate protein(s), if any; (iii) a Greek letter to identify non-vertebrate paralogs, if any and (iv) a three-letter species code. iGluRs from *A. thaliana* were used as an outgroup. All information on species and proteins used is given in *Figure 1—source data 2*. Phylogenetic reconstruction was performed using Bayesian inference. The amino acid substitution model used was Vt + G + F, number of generations: 14269000, final standard deviation: 0.007016 and potential scale reduction factor (PSRF): 1.000. Scale bar denotes number of amino acid substitutions per site. Although the GluAkdf2_Tad protein localizes to the Delta class in this tree, we do not consider this molecule as a confident member of this class. This is because the statistical support provided by the Bayesian analysis is low and because the Maximum-likelihood analysis (see *Figure 1—figure supplement 1*) does not position this protein in the Delta branch.

DOI: https://doi.org/10.7554/eLife.35774.003

The following source data and figure supplements are available for figure 1:

**Source data 1.** Conservation of protein domains in ionotropic glutamate receptors from unreported groups.
DOI: https://doi.org/10.7554/eLife.35774.009
**Source data 2.** Reference table of species and proteins used in the phylogenetic analysis of iGluRs.
DOI: https://doi.org/10.7554/eLife.35774.010
**Source data 3.** Aligned protein sequences used to construct ionotropic glutamate receptor phylogenies.
DOI: https://doi.org/10.7554/eLife.35774.011
**Source data 4.** Table with MolProbity scores of 3D models.
DOI: https://doi.org/10.7554/eLife.35774.012
**Figure supplement 1.** Maximum-likelihood phylogeny of metazoan ionotropic glutamate receptors.
DOI: https://doi.org/10.7554/eLife.35774.004
**Figure supplement 2.** Multiple protein alignment of transmembrane regions M1, M3 and M4 from unreported iGluRs.
DOI: https://doi.org/10.7554/eLife.35774.005
**Figure supplement 3.** Three-dimensional models of Epsilon class members.
DOI: https://doi.org/10.7554/eLife.35774.006
**Figure supplement 4.** Multiple protein alignment of the M1-M2 intracellular loop and the Q/R and +4 sites.
DOI: https://doi.org/10.7554/eLife.35774.007
**Figure supplement 5.** Multiple protein alignment of iGluR residues involved in ligand-binding.
DOI: https://doi.org/10.7554/eLife.35774.008

more restricted. Deltas are almost completely absent from ecdysozoan species, as we could only find a single member of this class in priapulids (*P. caudatus*) and none in arthropods or nematodes. Similarly, Deltas are poorly represented in mollusks and, with the available data, absent in annelids. Finally, we could only identify Phi proteins in cephalochordates, hemichordates and echinoderms, indicating that this class might be lost in the lineages of protostomes and olfactores (i.e. vertebrates and urochordates). The AKDF subfamily also includes proteins from the non-bilateral phyla of porifera, placozoa and cnidarian. The exact organization of these proteins into classes is not as straightforward as for bilateral proteins. The Bayesian and ML analysis only agree in the position of 12 iGluRs from the sponge *O. carmela,* these would constitute the only clear class in non-bilaterals, which we have termed AKDF-Oca.

Another example of a multiple lineage-specific event that occurred during animal evolution of iGluRs can be observed in the evolution of AMPA and Kainate proteins among protostomes. The general iGluRs phylogeny (*Figure 1*) suggests that ecdysozoan species have expanded their repertoire of Kainate subunits when compared with lophotrochozoans (e.g. mollusks, annelids), since *C. teleta* and *L. gigantea* only presents one and two genes coding for Kainate receptors, respectively. Contrarily, we found more AMPA subunits in lophotrochozoans than in ecdysozoan species. To investigate whether the two protostome lineages have alternatively expanded genes coding for AMPA or Kainate subunits we conducted a phylogenetic analysis of these two classes using eight species of ecdysozoans and seven of lophotrochozoans with well-characterized genomes (*Figure 3*

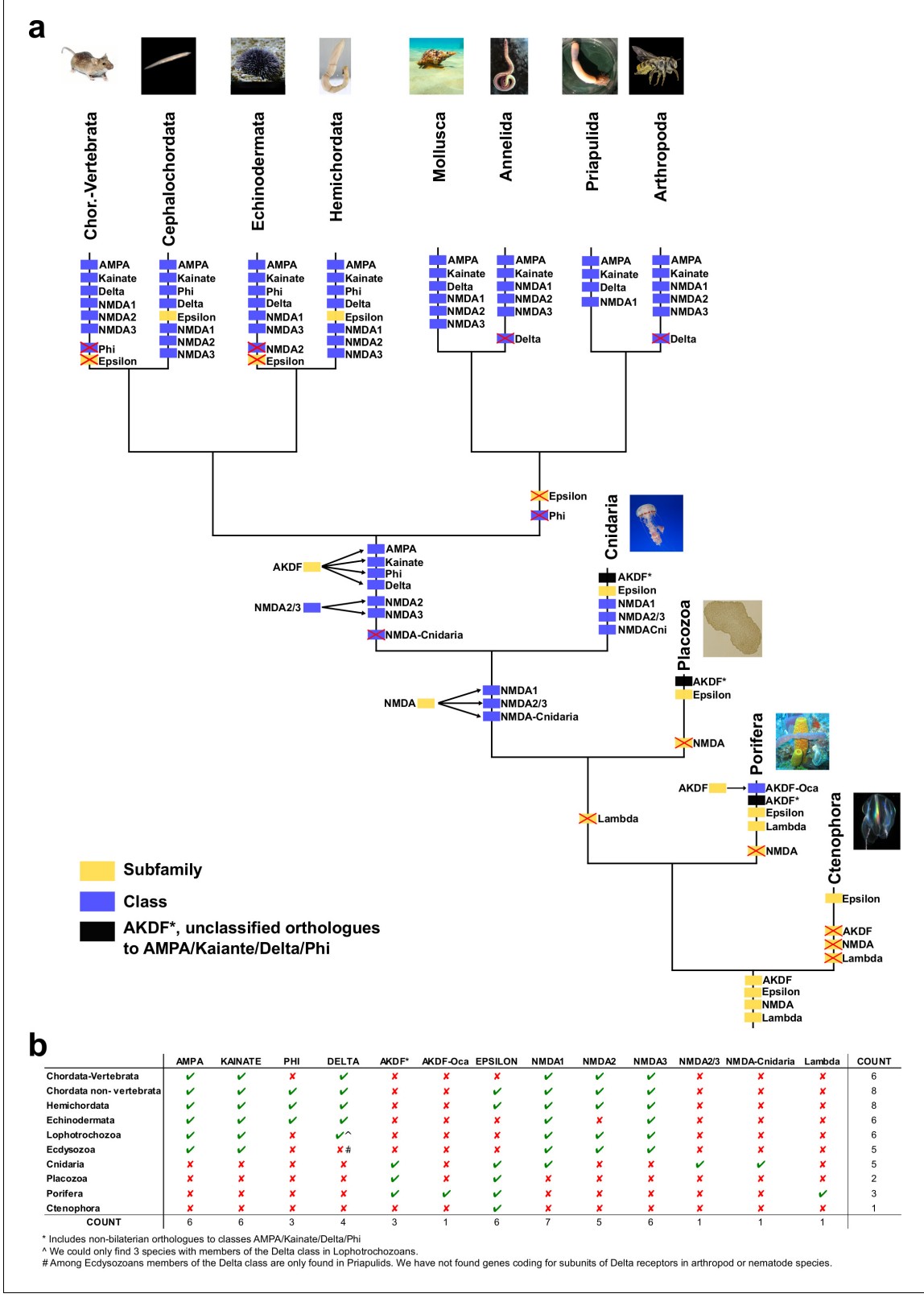

**Figure 2.** Schematic representation of iGluRs metazoan evolution. (a) Summary tree showing the evolution of iGluR subfamilies and classes in the metazoan lineages investigated. Each branch corresponds with one lineage. Phylogenetic subfamilies are represented by yellow boxes and classes by blue boxes. The four subfamilies present in the ancestor of all current metazoan lineages are shown at the base of the tree. Duplications of subfamilies in ancestors of current lineages are indicated. When a class or subfamily is lost in a lineage or in an ancestor, the corresponding box is crossed out with

*Figure 2 continued on next page*

*Figure 2 continued*

a red cross. Mollusca and annelida are lophotrochozoans and priapulida and arthropoda ecdysozoans. In priapulida NMDA2s and NMDA3s were not investigated. (**b**) Table indicating the presence or absence of iGluR subfamilies and classes in the metazoan lineages investigated. When a phylogenetic group is present in a lineage it is indicated by a green tick and if it is absent by a red cross. The last column shows the total number of groups found in each phylum. The last row shows the number of phyla where each phylogenetic group is present.

Image credit: Placozoa, author Oliver Voigt, licensed under CC BY-SA 3.0 Germany license; source https://commons.wikimedia.org/wiki/File:Trichoplax_mic.jpg; *P caudatus*, author Shunkina Ksenia, licensed under CC BY 3.0 source https://commons.wikimedia.org/wiki/File:Priapulus_caudatus.jpg; Hemichordata, released under GNU Free Documentation License, source https://commons.wikimedia.org/wiki/File:Eichelwurm.jpg; Cephalochordata, author Hans Hillewaert, licensed CC BY-SA 4.0 International license, source https://commons.wikimedia.org/wiki/File:Branchiostoma_lanceolatum.jpg.

DOI: https://doi.org/10.7554/eLife.35774.013

and *Figure 3—figure supplement 1*). Nematodes were left out of the analysis as they lack Kainate receptors (*Brockie et al., 2001*). This analysis retrieved 40 lophotrochozoan genes coding for AMPA subunits but only 15 coding for Kainates. The opposite scenario was observed in the genomes of ecdysozoan species, with 10 AMAP and 40 Kainate proteins,. Yet, among ecdysozoans the priapulid *P. caudatus has* two AMPA and two Kainate subunits, indicating that the expansion of Kainate receptors might be exclusive to arthropods. Overall the AMPA:Kainate ratio resulted to be around 1:4 in ecdysozoans and 4:1 in lophotrochozoans.

## Sequence conservation and ligand specificity of unreported iGluR phylogenetic groups

All proteins from unreported groups (i.e. subfamilies and classes) present well-conserved sequences in iGluR domains, including transmembrane domains or residues involved in receptor tetramerization (*Figure 1—figure supplement 2* and *Figure 1—source data 1*). Three-dimensional (3D) models of two Epsilon subunits from amphioxus (GluE1 and GluE7) indicate that their general fold is well preserved (*Figure 1—figure supplement 3a*). The only noticeable distinction in proteins from these groups is an insertion in the intracellular loop between the first and second transmembrane domains in Epsilon proteins. This insertion is particularly distinct in ctenophore iGluRs, having been termed as the cysteine-rich loop (*Alberstein et al., 2015*) (*Figure 1—figure supplement 4*). We have also identified a sequence difference among Epsilon proteins. Ctenophore iGluRs have two cysteines that form a disulfide bond at loop 1 of the ligand binding domain (*Alberstein et al., 2015*), which are also present in NMDA proteins. Nevertheless, this element is absent from the remaining members of the Epsilon subfamily.

The 'SYTANLAAF' motif, essential for channel gating (*Traynelis et al., 2010*), is also well conserved in most sequences, in particular the second, fourth and fifth residues (*Figure 1—figure supplement 2*). Nevertheless, all members of the Lambda subfamily and some proteins of the Phi class present lower levels of conservation in this sequence. Whether these changes have a functional impact is something that will require further investigation. The Q/R site (Q586, residue numbering according to mature rat GluA2) and the acidic residue located four positions downstream D/E590 (*Figure 1—figure supplement 4*) are involved in calcium permeability and polyamine block of AMPA and Kainate receptors (*Bowie and Mayer, 1995*; *Koh et al., 1995*; *Kamboj et al., 1995*). Of these two positions the latter is much better conserved, especially outside ctenophores and the Lambda subfamily. We have identified an acidic residue at position 590 in 84 out of 122 iGluRs from unreported groups, including cnidarian NMDAs. Yet, only 1/3 of these proteins present a glutamine (Q) at position 586. This includes most AKDFs and Epsilon proteins from non-ctenophores, contrarily, none of the Phi subunits presents a Q586.

The key ligand binding residues involved in fixing the amino acid backbone ($\alpha$−amino and $\alpha$−carboxyl) are Arg485 and an acidic residue at position 705 (*Naur et al., 2007*; *Armstrong and Gouaux, 2000*; *Mayer, 2005*; *Furukawa et al., 2005*; *Yao et al., 2008*). These two positions are well conserved in 94 of the 122 proteins from unreported groups, suggesting that their endogenous ligand is an amino acid (see *Figure 1—figure supplement 3b* for a 3D representation of ligand binding by GluE1 and *Figure 1—figure supplement 5* for an alignment of iGluR residues involved in ligand binding). The residue changes found in the remaining 28 proteins would render them unable to bind an amino acid (*Figure 1—figure supplement 5*). This is are particularly common among class Phi proteins from amphioxus and in NMDA-Cnidaria.

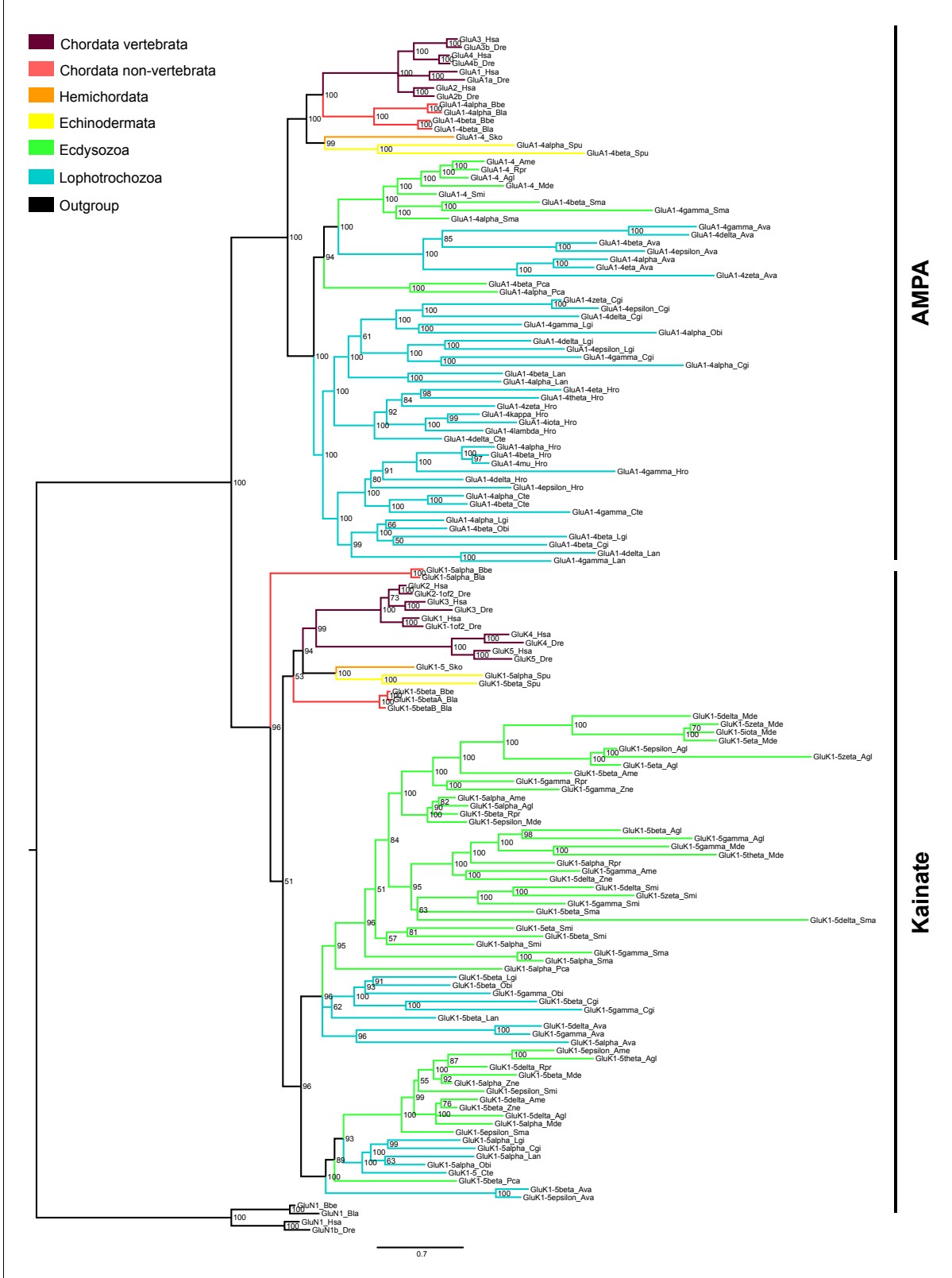

**Figure 3.** Bayesian phylogeny of AMPA and Kainate classes in protostomes. Ionotropic glutamate receptors classes are indicated at the right. Posterior probabilities are shown at tree nodes and protein names at the end of each branch. Tree branches are colored based on phylum, as indicated in the legend. Protein names from non-vertebrate species are composed of four parts: (i) 'GluR#', where # is a one letter code denoting class (A for AMPA and K for Kainate); (ii) a number, or range of numbers, denoting orthologous vertebrate protein(s), if any; (iii) a Greek letter to identify non-vertebrate

*Figure 3 continued on next page*

*Figure 3 continued*

paralogues, if any and (iv) a three-letter species code. GluN1s from *chordates* were used as an outgroup. All information on species and proteins used in this phylogeny is given in *Figure 3—source data 2*. Phylogenetic reconstruction was performed using Bayesian inference. The amino acid substitution model used was Vt + I + G, number of generations: 8868000, final standard deviation: 0.0072 and potential scale reduction factor (PSRF): 1.001. Scale bar denotes number of amino acid substitutions per site.

DOI: https://doi.org/10.7554/eLife.35774.014

The following source data and figure supplement are available for figure 3:

**Source data 1.** Aligned protein sequences used to construct AMPA and Kainate class phylogenies in protostomes.
DOI: https://doi.org/10.7554/eLife.35774.016
**Source data 2.** Reference table of species and proteins used in the phylogenetic analysis of AMPA and Kainate classes in protostomes.
DOI: https://doi.org/10.7554/eLife.35774.017
**Figure supplement 1.** Maximum-likelihood phylogeny of AMPA and Kainate classes in protostomes.
DOI: https://doi.org/10.7554/eLife.35774.015

Residues involved in ligand selectivity show higher variability. These are located at positions 653 and 655, and are occupied by glycine and threonine in glutamate-binding proteins and by serine and a non-polar residue in glycine-binding iGluRs. However, a recent study of ctenophore receptors has found that position 653 can be occupied by serine or threonine in glutamate-binding iGluRs, and by an arginine in glycine-binding subunits (*Alberstein et al., 2015*). Based on this previous knowledge we have predicted the ligand specificities of most previously unreported receptors. The preferred ligand could be confidently predicted for 72 out of the 94 proteins with well-conserved residues involved in fixing the amino acid backbone.

Interestingly, all unreported groups comprise glycine- and glutamate-specific iGluRs. Gly-specific receptors slightly outnumber those predicted to respond to glutamate (overall ratio about 3:2). The Lambda subfamily would include three proteins specific for glutamate and one for glycine, while seven remain with an unknown selectivity. Of note, the protein predicted to bind glycine (Glu-L5_Oca) displays an arginine at position 653, a feature which had only been reported in ctenophores (*Alberstein et al., 2015*). This residue would form a salt bridge with Glu423, which is key for glycine selectivity in ctenophores (*Alberstein et al., 2015*). Most Epsilon and AKDF proteins would preferably bind glycine, although ctenophores present a similar number of Epsilon receptors predicted to respond to glycine or glutamate (*Figure 1*) (*Alberstein et al., 2015*). In the Phi class we also found a similar number of receptors binding glycine and glutamate. Finally, we could only predict binding specificity for two of the 9 NMDA-Cnidaria proteins, as they present many changes in the residues involved in either amino acid backbone binding or side chain recognition.

Interestingly, the 22 proteins for which we could not confidently predict their ligand selectivity (*Figure 1—figure supplement 5*), present a limited number of residues occupying position 653 and 655, suggesting constrained evolution. Of these: (i) nine present residues with negative polarity at both positions, being candidates to bind glutamate, (ii) six present a Gly653 and a non-polar residue at position 655, and thus are candidates to bind glycine, (iii) five proteins, all from the *Branchiostoma* genus, present a tyrosine at position 653. A structural model of one of these receptors, GluE7 (*Figure 1—figure supplement 3c*), shows that a Tyr653 aromatic side chain would occupy the ligand-binding pocket, strongly suggesting that amino acid binding would be blocked. Finally, (iv) two proteins present a phenylalanine in either of the two positions and remain unclassified.

## Epsilon and Phi iGluR proteins are highly expressed in the nervous system and traffic to the plasma membrane

We used quantitative PCR (qPCR) to investigate gene expression levels of all iGluR subunits identified in *B. lanceolatum,* including those from the Epsilon and Phi groups. All 24 *B. lanceolatum* iGluR subunits identified in silico were found expressed in amphioxus, with the exception of *Grie5* (*Figure 4a*). Furthermore, they all showed a significantly higher expression in the nerve cord as compared to the whole body, suggesting tissue-enriched expression. While we observed low expression levels for Epsilon genes coding for subunits with a tyrosine at position 653 (*Grie5-8*), which according to the 3D model would block the ligand-binding pocket, the expression of Grif1-2, also presenting the same tyrosine, reach much higher levels, comparable to those of subunits from the Kainate,

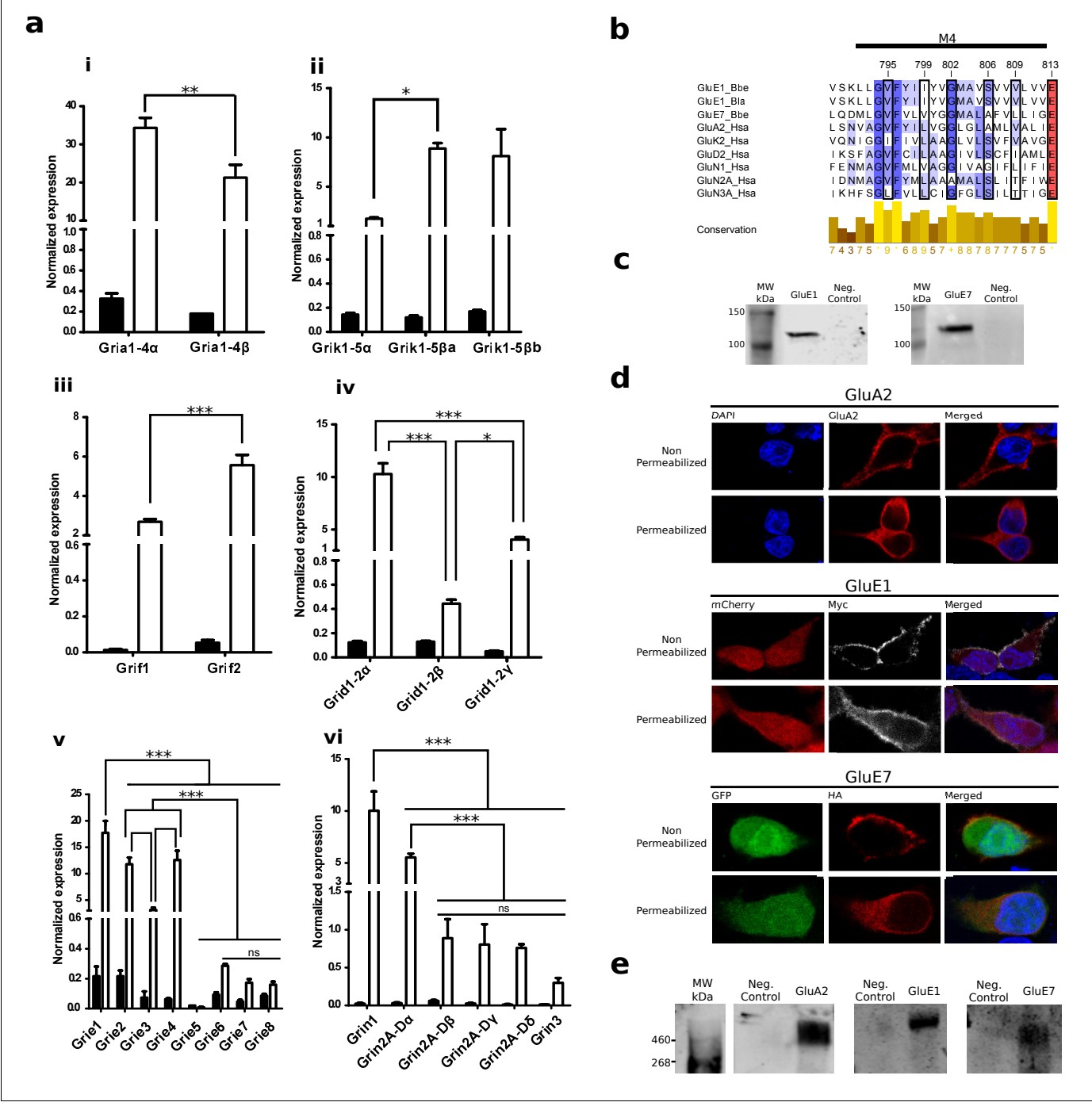

**Figure 4.** Expression and functional analysis of amphioxus iGluRs. (**a**) iGluRs mRNA expression (mean and standard deviation) in *Branchiostoma lanceolatum*. Bars show average relative expression of *B. lanceolatum* (amphioxus) iGluR genes as determined by qPCR. Filled bars represent whole body and open bars nerve cord expression levels. Note that all genes show significantly enriched expression in the nerve chord relative to the whole body, with the exception of *Grie5* (Student's t-test, n = 3). Expression level in the nerve chord is compared across genes of the same class. Statistics: pair comparisons were done by Student's t-test, n = 3, multiple comparisons were done by one-way ANOVA followed by Tukey's Post-Hoc test, n = 3. Significance levels: ***p < 0.001, **p < 0.01 and *p < 0.05; ns, not significant. (i) AMPA class. (ii) Kainate class. (iii) Phi class. (iv) Delta class. (v) Epsilon subfamily. (vi) NMDA classes. (**b**) Multiple sequence alignment of iGluRs transmembrane region M4 containing residues involved in tetramerization, these are indicated by a black frame. Higher amino acid conservation is represented by increasing intensity of blue background and by a bar chart at the bottom. Sequences included are GluE1 and GluE7 from amphioxus and representatives of human iGluRs. (**c**) Immunoblot of chimeric GluE1 and GluE7, containing the signal peptide from rat GluA2, expressed in HEK293T cells. Proteins were detected using the immuno-tags (c-Myc and HA, respectively) located after the rat signal peptide. Protein extracts from non-transfected cells were loaded as negative controls. (**d**) Immunofluorescence

*Figure 4 continued on next page*

*Figure 4 continued*

of HEK293T cells expressing rat GluA2 (top), cMyc-tagged GluE1 (middle) or HA-tagged GluE7 (bottom). Both non-permeabilized and permeabilized conditions are shown. (**e**) Immunoblot of tetrameric rat GluA2, GluE1 and GluE7 expressed in HEK293T cells. Amphioxus proteins were detected using the immuno-tags (c-Myc and HA, respectively) located at the N-terminus of each sequence. Protein extracts from non-transfected cells were loaded as negative controls.

DOI: https://doi.org/10.7554/eLife.35774.018

The following source data and figure supplements are available for figure 4:

**Source data 1.** qPCR values used to generate *Figure 4a*.

DOI: https://doi.org/10.7554/eLife.35774.021

**Figure supplement 1.** Wild-type GluE1 and GluE7 expression in HEK293T cells and genetic strategy used to add a signal peptide.

DOI: https://doi.org/10.7554/eLife.35774.019

**Figure supplement 2.** List of primers used in qPCR experiments.

DOI: https://doi.org/10.7554/eLife.35774.020

Delta or NMDA classes. Thus, the presence of a tyrosine at position 653 does not appear to be directly correlated with low expression levels.

Amphioxus genes coding for GluE1 and GluE7 were synthesized in vitro and transiently expressed in HEK293T cells for functional studies. Wild-type GluE1 and GluE7, which are not predicted to have a canonical signal peptide by SignalP 4.1 (*Nielsen, 2017*), expressed well but were not trafficked to the plasma membrane (*Figure 4—figure supplement 1a–d*), even though residues involved in tetramerization (*Salussolia et al., 2013*) are well conserved (*Figure 4b*). We thus synthesized new variants of these genes with the signal peptide from rat GluA2 (*Figure 1—figure supplement 1cd*). These constructs also expressed well (*Figure 4c*) and now were efficiently trafficked to the plasma membrane, as indicated by the staining observed in non-permeabilized cells (*Figure 4d*). Furthermore, analysis of receptor oligomerization, performed using non-denaturing gel electrophoresis and immunoblot, clearly indicates that both proteins form homotetramers in vitro (*Figure 4e*).

## Ligand specificity and electrophysiological properties of Epsilon proteins from amphioxus

We next investigated the gating properties of two Epsilon proteins from amphioxus, GluE1 and GluE7. The presence of a serine and a tryptophan at positions 653 and 704, respectively, suggested that GluE1 would bind glycine. Indeed, neither glutamate nor aspartate elicited a response in our experimental settings. Instead, glycine application was able to elicit an inward whole-cell current at a membrane potential of −60 mV (*Figure 5a*). Interestingly, the chemically related amino acids alanine and D-serine only generated very low responses, indicating a high selectivity of the GluE1 homotetramer for glycine.

The Epsilon receptor displayed a strong inward rectification, even in the absence of added polyamines in the intracellular solution (*Figure 5b,c*). This behavior is characteristic of unedited AMPA and Kainate receptors displaying a glutamine (Q) and an acidic residue at positions 586 and 590, respectively (*Bowie and Mayer, 1995*; *Koh et al., 1995*; *Kamboj et al., 1995*) and GluE1 presents a glutamine and an aspartic acid at these positions (*Figure 1—figure supplement 4*). Glycine-mediated currents showed a slow rate of recovery from desensitization when compared with AMPA or Kainate mammalian receptors, requiring 20–25 seconds until a complete recovery was achieved and a full response of the same magnitude could be recorded (*Figure 5d,e*). Similar observations have been made with ctenophore receptors activated by glycine in which the recovery from desensitization has an unusually long time constant of 81 seconds (*Alberstein et al., 2015*).

Finally, functional studies on receptors formed by GluE7 did not retrieve any positive results. None of the following amino acids: glutamate, aspartate, asparagine, glycine, alanine or D-serine elicited a response in our experimental system. We hypothesize that, as predicted by the 3D model, the presence of a tyrosine at position 653 renders a homomeric form of this receptor unable to function as an amino acid-gated ion channel.

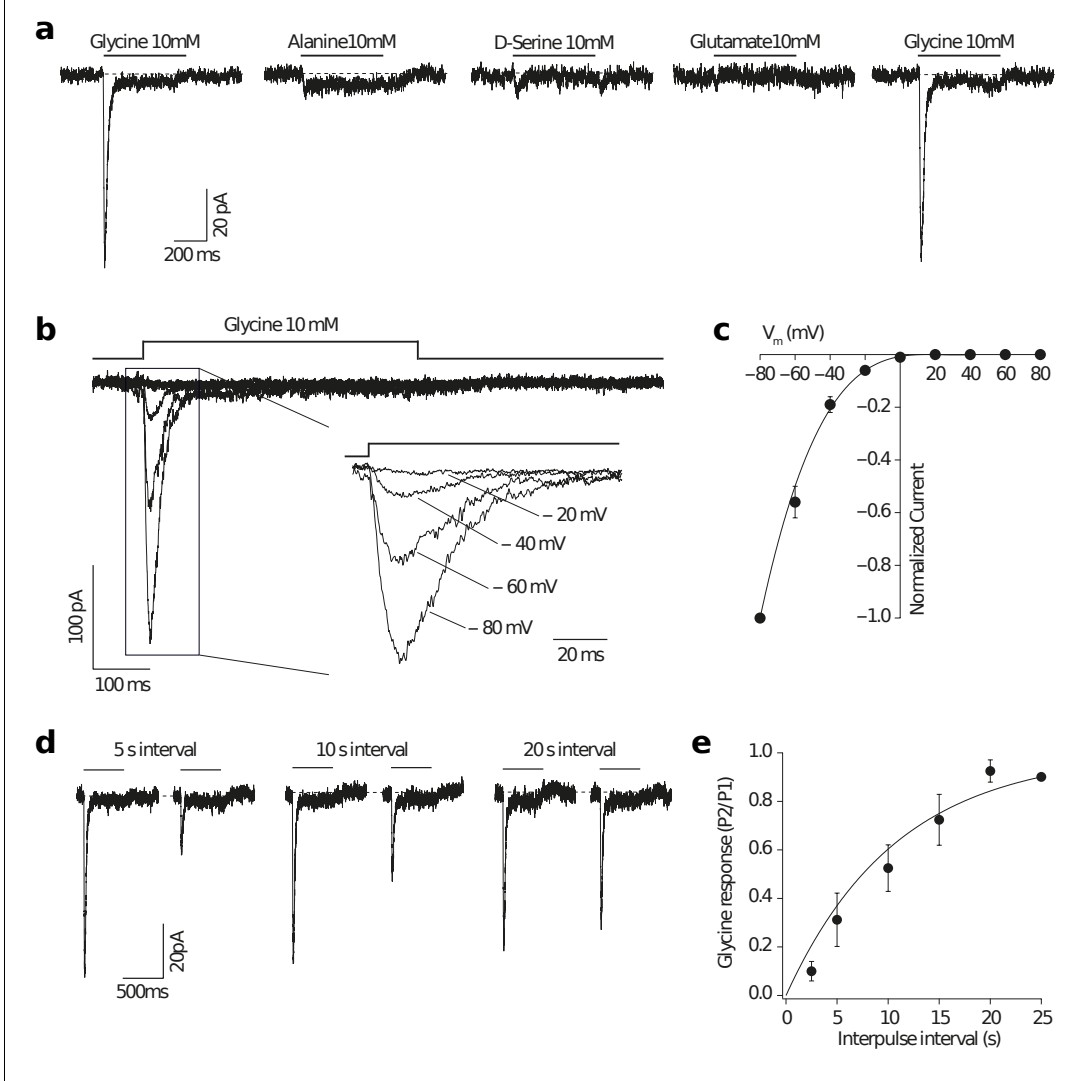

**Figure 5.** Glycine activates an amphioxus homomeric Epsilon receptor. (a) Representative homomeric GluE1 (from *B. lanceolatum*) whole-cell currents evoked by a rapid pulse (500 ms) of different amino acids (10 mM) in HEK293T cells. Left and right glycine-mediated currents denote agonist application before and after alanine, D-serine and glutamate applications respectively for ruling out run-down of the currents. (b) Representative GluE1 responses to 10 mM glycine at different membrane voltages (from −80 to +80 mV in 20 mV steps). Note that a strong inward rectification can be observed even in the absence of added polyamines in the intracellular solution. Inset: currents at negative membrane voltages are shown. (c) Current-voltage relationship for peak currents evoked by glycine (500 ms, 10 mM) applied to whole HEK293T cells containing homomeric GluE1 subunits normalized for the current at −80 mV (n = 3) fitted to a 5th order polynomial function. Error bars represent SEM. (d) Homomeric GluE1 glycine-mediated currents recorded at different time intervals by using a paired pulse protocol. (e) Rate of recovery of desensitization fitted to a single exponential of time constant 10.8 s (n = 3–5). Plot shows the average ratio values (P2/P1) and SEM (error bars).

DOI: https://doi.org/10.7554/eLife.35774.022

## Phylogenetics of metazoan metabotropic glutamate receptors reveals a sister group of classes I to III

We next performed a phylogenetic study of metabotropic glutamate receptors (*Figure 6* and *Figure 6—figure supplement 1*). This analysis has revealed that the three historical mGluR classes (I to III) have a sister group. Following the current nomenclature we have named this as class IV. The existence of this class had already been proposed on the bases of three insect proteins (*Mitri et al., 2004*). Yet, here we show that this class is actually present in all bilateral phyla, excluding vertebrates. Furthermore, we also show that class IV appeared together with classes I-III before radiation of bilateral lineages. We have identified clear orthologues to class I-IV in porifers, placozoans and

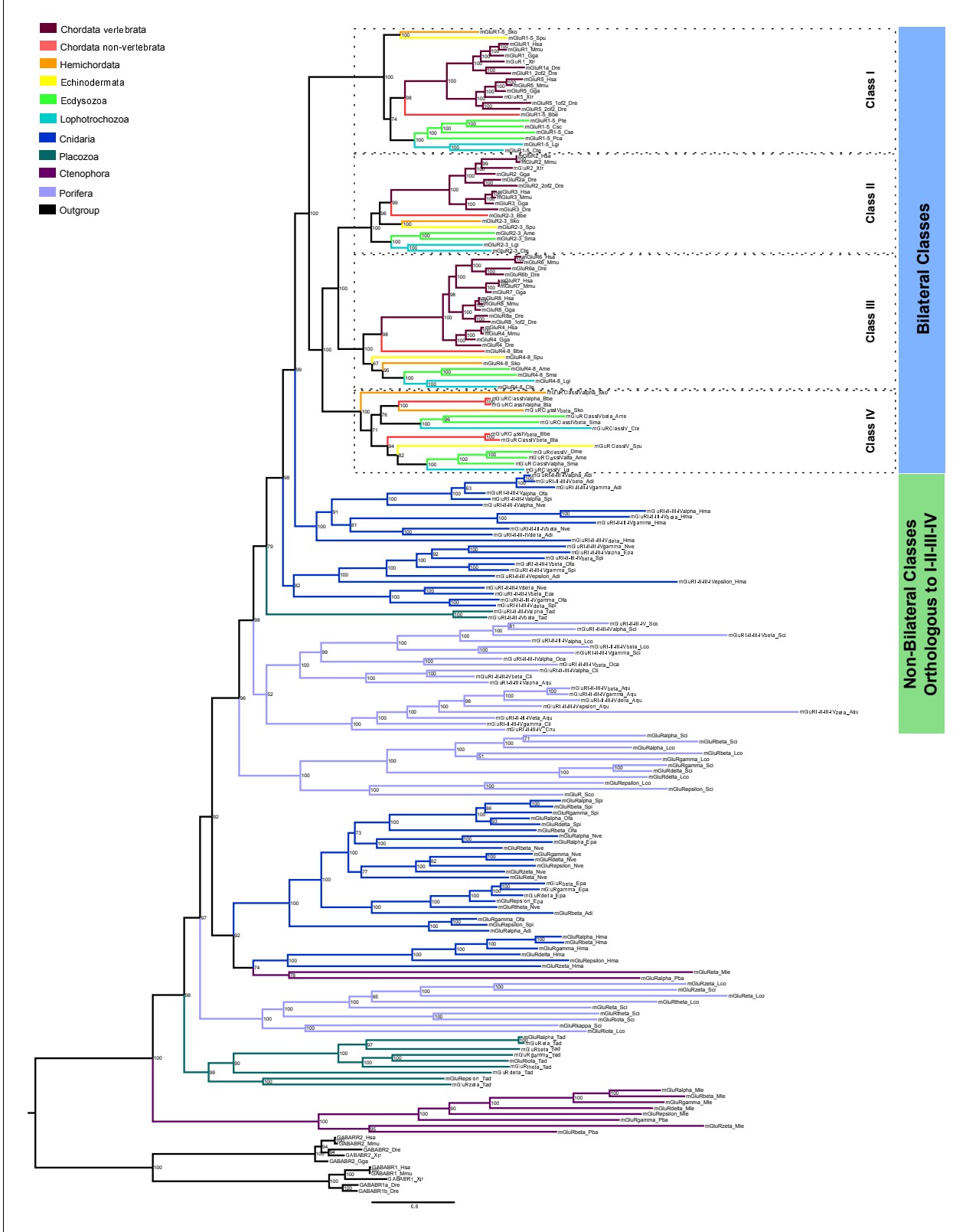

**Figure 6.** Bayesian phylogeny of metazoan metabotropic glutamate receptors. Identified metabotropic glutamate receptor classes from bilateral and non-bilateral organisms are indicated by colored boxes at the right. Dashed boxes further highlight individual classes from bilateral organism. Posterior probabilities are shown at tree nodes and protein names at the end of each branch. Tree branches are colored based on phylum, as indicated in the legend. Protein names from non-vertebrate species are composed of four parts: (i) 'mGluR', followed by a number, or range of numbers, denoting

*Figure 6 continued on next page*

*Figure 6 continued*

orthologous vertebrate protein(s), if any (for Class IV and group I-II-III-IV proteins, the name is followed by the name of the class/group); (ii) a Greek letter to identify non-vertebrate paralogs, if any and (iv) a three-letter species code. GABA-B receptors from vertebrates were used as an outgroup. All information on species and proteins used in this phylogeny is given in *Figure 6—source data 2*. Phylogenetic reconstruction was performed using Bayesian inference. The amino acid substitution model used was WAG + I + G + F, number of generations: 5327000, final standard deviation: 0.004788 and potential scale reduction factor (PSRF): 1.001. Scale bar denotes number of amino acid substitutions per site.

DOI: https://doi.org/10.7554/eLife.35774.023

The following source data and figure supplements are available for figure 6:

**Source data 1.** Conservation of protein domains in metabotropic glutamate receptors from unreported classes.

DOI: https://doi.org/10.7554/eLife.35774.028

**Source data 2.** Reference table of species and proteins used in the phylogenetic analysis of mGluRs.

DOI: https://doi.org/10.7554/eLife.35774.029

**Source data 3.** Aligned protein sequences used to construct metabotropic glutamate receptor phylogenies.

DOI: https://doi.org/10.7554/eLife.35774.030

**Source data 4.** qPCR values used to generate *Figure 6—figure supplement 2b*.

DOI: https://doi.org/10.7554/eLife.35774.031

**Figure supplement 1.** Maximum-likelihood phylogeny of metazoan metabotropic glutamate receptors.

DOI: https://doi.org/10.7554/eLife.35774.024

**Figure supplement 2.** Multiple protein alignment of mGluR residues involved in ligand binding and expression levels of *B.lanceolatum* mGluR genes.

DOI: https://doi.org/10.7554/eLife.35774.025

**Figure supplement 3.** Multiple protein alignment of mGluR transmembrane regions.

DOI: https://doi.org/10.7554/eLife.35774.026

**Figure supplement 4.** List of primers used in qPCR experiments.

DOI: https://doi.org/10.7554/eLife.35774.027

---

cnidarians but not in ctenophores. These are organized into four classes, two from cnidarians, and one from placozoans and porifers (*Figure 6*). We have also identified non-bilaterian mGluRs that fall outside the above-mentioned classes. Unfortunately, the Bayesian and ML phylogenies do not agree on the exact organization of these early divergent mGluRs, except for the fact that they diverge prior to bilaterian classes. For this reason we have left these sequences unclassified. Whether these sequences belong to one, or even multiple classes that would have been lost in bilateral organisms is something that will require further investigation.

Although all class IV proteins show well conserved sequences overall (*Figure 6—figure supplement 2a*, *Figure 6—figure supplement 3* and *Figure 6—source data 1*), two residues critical for glutamate binding, Arg78 and Lys409, are non-conservatively replaced by non-polar or acidic residues in all class IV proteins identified (*Figure 6—figure supplement 2a*, residue numbering corresponds to human mGluR1). These changes are predicted to hamper glutamate binding and, indeed, functional studies of a class IV receptor from fruit fly indicated that it does not respond to this amino acid (*Mitri et al., 2004*). All class IV proteins would share this feature. On the other hand, residues involved in contacts with the amino acid backbone are well conserved (*Figure 6—figure supplement 2a*), suggesting that these proteins might bind an amino acid other than glutamate. Similarly, mGluR residues from most non-bilaterian sequences involved in binding the amino acid backbone are highly conserved. Among non-bilaterian proteins the residues involved in glutamate binding are only conserved in approximately half of the proteins from classes orthologous to I-II-III-IV. Finally, we investigated mGluRs expression in amphioxus following the same procedure described for iGluRs. All five amphioxus mGluRs showed an enriched expression in the nerve cord, including the two class IV genes. Noticeably, these two genes showed significantly higher expression levels than orthologues of vertebrate classes (*Figure 6—figure supplement 2b*).

## Discussion

We have performed what to our knowledge is the most comprehensive phylogenetic study of metazoan glutamate receptors. This has revealed that their evolutionary history is much more complex than what is currently acknowledged, especially for the family of iGluRs. Our study has also revealed the existence of unreported phylogenetic groups in both ionotropic and metabotropic glutamate

receptors. Importantly, our data indicate that the evolution of glutamate receptors has not occurred in an unequivocal incremental manner only in those clades with more elaborated neural systems, but it has rather followed an scattered lineage-specific evolutionary history. This means that certain lineages have experienced the gain, loss, expansion or reduction of specific phylogenetic groups.

Our phylogenetic analysis indicates that the family of iGluRs is actually divided into four unreported subfamilies that we have termed Lambda, Epsilon, NMDA and AKDF. Interestingly, this general organization was already present in the last common ancestor of all metazoans and later duplications within NMDA and AKDF subfamilies resulted in the formation of well-known iGluR classes. The other two subfamilies are absent from the majority of model species used in neuroscience research. The NMDA subfamily diversified into classes NMDA1-3 but also into the NMDA2/3 and NMDA-Cnidaria. Similarly, the AKDF subfamily diversified into the AMPA, Kainate and Delta classes, but also into the previously unreported Phi class. We have also identified and AKDF class exclusive to porifers, represented by sequences form *O. carmela*. Most well-studied iGluR classes are the result of duplications in ancestors of current bilateral species, >650 million years ago (mya) (*Kumar et al., 2017*), only class NMDA1 originated earlier, as cnidarians present members within this class. The Epsilon subfamily, which includes all iGluRs from ctenophores, is the only subfamily present in all non-bilateral phyla investigated, including sponges. It is thus the subfamily presenting a larger phylogenetic spread, as it is also present in hemichordates and in non-vertebrate chordates. On the other hand, the unreported Phi class shows a more restricted phylogenetic spread, as it is present only in three deuterostome phyla. Moreover, Lambda proteins seem restricted to Porifers, which constitutes an interesting evolutionary case due to maintenance of a glutamate receptor family in a phylum without nervous system.

The phylogenetic analysis of metabotropic glutamate receptors has allowed us to unambiguously establish the existence of a sister group to the well-known classes I, II and III. Following the present nomenclature we have named this as class IV. This class had been previously proposed based on the identification of three insect mGluRs that did not cluster with members of known classes (*Mitri et al., 2004*). Here we show that class IV is not restricted to insects, but is actually present in all bilaterian phyla investigated, with the exception of vertebrates where this class has been lost. Interestingly, as it occurs for most well-known iGluR classes, mGluR classes I-IV appeared simultaneously in the ancestor of bilaterals. Our phylogenetic analysis also indicates that the non-bilateral phyla of cnidarians, placozoans and porifers present clear orthologues to classes I-IV, which are organized into four classes, while we failed to find any in the early-branching ctenophores. Finally, we were unable to confidently classify many non-bilateral mGluRs, which might constitute one or more classes.

We have identified many examples of lineage-specific evolutionary events. These would antagonize with a model in which species with less elaborated nervous systems would present GluR families with lower complexity. The most noticeable examples are: (i) the absence of all subfamilies but Epsilon in analyzed ctenophores, (ii) the loss of Delta receptors from arthropods, nematodes and annelid species investigated, (iii) the loss of the Epsilon subfamily in vertebrates, echinoderms and protostomes, (iv) the loss of the Phi class in vertebrates and studied protostomes, (v) the specific expansion of Kainate receptors in arthropods, which contrasts with the expansion of AMPA receptors in its sister lineages of mollusks and annelids, (vi) the large expansion of the Epsilon subfamily in ctenophores, placozoans and cephalochordates and, finally (vii) the loss of mGluR class IV in vertebrates.

Along the same line, it is interesting to note that amphioxus (*B. belcheri and B. lanceolatum*), with a simple nervous system, have over 20 genes encoding iGluRs, while mammals have 18. Other non-vertebrate species also present large numbers of iGluRs, including the 19 iGluRs identified in the sponge *O. carmela* or the 17 present in the ctenophore *M. leidyi,* to mention a few. Similarly, the cnidarian *A. digitifera* and the ctenophore *M. leidyi* have seven mGluRs each, while the placozoan *T. adhaerens* presents eleven, three more than the eight mGluRs found in the human genome. The large number of GluRs found in many non-vertebrate animals suggests that there has been an evolutionary trend to increase their number in many metazoan lineages.

Our experimental results suggest that unreported receptors would play a role in the nervous system, as Epsilon, Phi and mGluR class IV genes are highly expressed in the nerve cord of amphioxus. Nevertheless, whether all these proteins are expressed at the synapse and act as neurotransmitter receptors is an issue that will require further investigation. Their presence in other tissues, such as sensory organs, cannot be ruled out. Those receptors showing more divergent sequences,

particularly in residues involved in ligand binding, might respond to other molecules. For instance, they could behave as chemoreceptors, as it is the case of antennal receptors found in insects (*Croset et al., 2010*; *Benton et al., 2009*).

Proteins from all unreported groups generally present a good conservation of residues involved in binding the amino acid backbone, indicating that their ligand would be an amino acid or a closely related molecule. Interestingly, we could identify proteins predicted to bind either glycine or glutamate in all unreported iGluR subfamilies and classes. If our functional predictions are correct, the ability to recognize one or the other amino acid would have emerged repeatedly in all unreported iGluR phylogenetic groups. Unexpectedly, the nature of the residues conferring amino acid specificity indicates that only a minority of proteins from unreported GluR groups would respond to glutamate. Sequence analysis and structural considerations strongly suggest that class IV mGluRs will not bind glutamate and that among non-bilateral mGluRs only a minority, belonging to classes orthologous to I-II-III-IV, are predicted to bind to this neurotransmitter. Similarly, among unreported iGluR groups, the number of proteins binding glycine outnumbers those binding glutamate. Interestingly, we report a glycine-binding poriferan protein (GluL5_Oca) with a structural feature that had only been reported in ctenophores (*Alberstein et al., 2015*). This is an Arg653 that through establishing a salt bridge with Glu423 confers glycine specificity (*Alberstein et al., 2015*). We thus report that this structural element is not exclusive to ctenophores. We have also identified iGluR subunits with important changes in critical ligand binding residues, indicating that they might have evolved new biological functions, for example, response to other, as yet unidentified small molecules.

The activation of Epsilon receptors by glycine has been experimentally corroborated by electrophysiological analysis of homotetrameric receptors composed by GluE7 from *M. leidy* (*Alberstein et al., 2015*) and GluE1 from amphioxus (this study). In our hands the amphioxus receptor showed a very high selectivity for glycine, since ion currents could not be elicited by chemically related amino acids such as serine or alanine. Glycine-binding Epsilon subunits from phyla other than ctenophores present structural features similar to those from glycine-binding iGluRs in vertebrates. The greater number of glycine receptors found in non-vertebrate species could be related to the higher abundance of this amino acid in their nerve cord as compared with the mammalian brain (*Pascual-Anaya and D'Aniello, 2006*).

Altogether, our phylogenetic analysis and experimental findings have uncovered the complex evolution of glutamate receptors within the metazoan kingdom. Our data indicate that the classification of iGluRs is not restricted to the six classes currently recognized. Instead, iGluRs are organized into four subfamilies: Lambda, Epsilon, NMDA and AKDF and ten classes with varying phylogenetic spread. With the data available, the NMDA subfamily is organized into classes NMDA1, NMDA 2, NMDA3, NMDA-Cnidaria and NMDA2/3, while subfamily AKDF contains classes AMPA, Kainate, Delta, Phi and AKDF-Oca. Both NMDA2/3 and AKDF-Oca are represented by sequences from only one species, further sequencing of non-bilateral species will be required to fully demonstrate their existence. Furthermore, the evolution of mGluRs has generated a sister group to classes I, II and III, class IV. We have also identified classes of non-bilaterian mGluRs orthologous to I-II-III-IV. We propose that the classification of these two families of GluRs, key to the physiology of the nervous system, has to be updated to include our findings.

## Materials and methods

**Key resources table**

| Reagent type (species) or resource | Designation | Source or reference | Identifiers | Additional information |
|---|---|---|---|---|
| Cell line (*Homo sapiens*) | HEK293T | American Type Culture Collection | Cat#: CRL-3216<br>RRID: CVCL_0063 | |
| Transfected construct (synthesize) | pIRES2_EGFP | Addgene | Cat. #: 6029–1 | |

*Continued on next page*

*Continued*

| Reagent type (species) or resource | Designation | Source or reference | Identifiers | Additional information |
|---|---|---|---|---|
| Transfected construct (synthesize) | pICherryNeo | Addgene | Cat. #: 52119 | |
| Transfected construct (synthesize) | *Grie1* in pICherryNeo | Invitrogen GeneArt Gene Synthesis | | |
| Transfected construct (synthesize) | *Grie7* in pIRES2_EGFP | Invitrogen GeneArt Gene Synthesis | | |
| Biological sample (*Branchiostoma lanceolatum*) | whole animal | | | Collected in the bay of Argelès-sur-Mer, France (latitude 42° 32′ 53′ N and longitude 3° 03′ 27′ E) |
| Biological sample (*Branchiostoma lanceolatum*) | nerve chord | | | Collected in the bay of Argelès-sur-Mer, France (latitude 42° 32′ 53′ N and longitude 3° 03′ 27′ E) |
| Antibody | Mouse anti-HA | Covance | Cat. #: MMS-101P RRID: AB_291259 | IF (1:200), WB (1:1000) |
| Antibody | Rabbit anti-c-Myc | Cell Signalling | Cat. #: 2272S RRID: AB_10692100 | IF (1:100), WB (1:1000) |
| Antibody | Mouse anti-GluA2 | Millipore | Cat. #: MAB397 RRID: AB_2113875 | IF (1:200), WB (1:1000) |
| Antibody | Alexa Fluor 555 donkey anti-mouse IgG | Invitrogen | Cat. #: A-31570 RRID: AB_2536180 | IF (1:1000) |
| Antibody | Alexa Fluor 647 goat anti-rabbit IgG | Life Technologies | Cat. #: A-21245 RRID: AB_2535813 | IF (1:500) |
| Antibody | Donkey anti-mouse | Li-cor | Cat. #: 926–32212 RRID: AB_621847 | WB (1:7500) |
| Antibody | Donkey anti-rabbit | Li-cor | Cat. #: 926–68073 RRID: AB_10954442 | WB (1:7500) |
| Recombinant DNA reagent | | | | |
| Sequence-based reagent | *Grie1* gene from B. Lanceolatum | | | |
| Sequence-based reagent | *Grie7* gene from B. Belcheri | | | |

*Continued on next page*

*Continued*

| Reagent type (species) or resource | Designation | Source or reference | Identifiers | Additional information |
|---|---|---|---|---|
| Sequence-based reagent | Seqeucne corresponding with rat *Gria2* signal peptide | | | |
| Chemical compound, drug | N-dodecyl-α-maltopyr anoside; DDM | Anatrace | Cat. #: D310HA | 2% w/v |
| Software, algorithm | pClamp10 | Molecular Devices | | |
| Software, algorithm | IgorPro | Wavemetrics | | |
| Software, algorithm | Neuromatic | doi: 10.3389/fninf.2018.00014 | RRID: SCR_004186 | |
| Software, algorithm | MrBayes 3.2.6 | doi: 10.1093/sysbio/sys029 | | |
| Software, algorithm | IQTree | doi: 10.1093/molbev/msu300 | | |
| Software, algorithm | MolProbity | doi: 10.1107/S09074 44909042073 | RRID: SCR_014226 | |
| Software, algorithm | MIFit | GitHub (*Smith, 2010*) | | |
| Software, algorithm | FIJI | doi: 10.1038/nmeth.2019 | RRID: SCR_002285 | |
| Other | CIPRES Science Gateway | doi: 10.1109/GCE.2010.5676129 | RRID: SCR_008439 | Free on-line super computing resource for evolutionary research |

## Identification of genes coding for members of glutamate receptor families in metazoan genomes

Phylogenetic analysis were performed with sequences from at least two species from each of the following metazoan phyla: Porifera, Ctenophora, Placozoa, Cnidaria, Lophotrochozoa, Ecdysozoa, Hemichordata, Chordata and Vertebrata, with the exception of placozoans for which only one species is available. When possible, we chose slowly evolving species. The complete lists of species used for iGluR phylogenies are given in *Figure 1—source data 2*. Species used in the phylogeny of metabotropic glutamate receptors are listed in *Figure 6—source data 2*. Sponge sequences were taken from (*Riesgo et al., 2014*), *B. lanceolatum sequences* were retrieved from unpublished genomic and transcriptomic databases (access was kindly provided by the Mediterranean Amphioxus Genome Consortium), *A. digitifera* and *P. flava* sequences were obtained from the Marine Genomics Unit (*Simakov et al., 2015*; *Shinzato et al., 2011*) and *P. bachei* sequences from NeuroBase (*Moroz et al., 2014*).

GluR sequences were identified using homology-based searches in a two-tier approach. Mouse glutamate receptors were used as search queries (iGluRs: Gria1-4; Grik1-5; Grid1-2, Grin1, Grin2A-D and Grin-3A-B; mGluRs: mGluR1-8). In a first search GluR homologs were identified using the BLASTP tool (*Altschul et al., 1990*) with default parameters. Subject sequences with an E-value below 0.05 were selected as candidate homologs. These were re-blasted against the NCBI database of 'non-redundant protein sequences' using the same BLAST tool. If the first hit obtained in the reciprocal BLAST was a glutamate receptor the sequence was included in the phylogenetic analysis. In a second stage the same mouse sequences were used to perform TBLASTN searches against

genomic and, when available, transcriptomic databases. Subject sequences not identified in the first tear and having an E-value below 0.05 were selected as candidate homologs. These were re-blasted using BLASTX against the NCBI 'non-redundant protein sequences' database. Finally, if the first hit of this search was a glutamate receptor the sequence was also included in the phylogenetic analysis. Identified iGluR sequences in which less than four residues of the SYTANLAAF motif (*Traynelis et al., 2010*) were conserved were not considered for the final phylogenetic analysis. mGluR sequences lacking two or more of the seven transmembrane regions were also discarded. The complete reference lists of all iGluRs used in the final phylogeny are given in files *Figure 1—source data 2*. The reference list of metabotropic glutamate receptors is presented in *Figure 6—source data 2*. The alignments used for the phylogenetic analysis of iGluRs, mGluRs and AMPAs and Kainates from protostomes are provided in *Figure 1—source data 3*, *Figure 3—source data 1* and *Figure 6—source data 3*.

## Phylogenetic analyses

The iGluR tree was constructed with 224 sequences identified in 26 non-vertebrate species (*Figure 1—source data 2*). The tree also included 18 iGluR sequences from vertebrates and two iGluR proteins from *A. thaliana*, used as an outgroup (*Chiu et al., 2002*). The phylogenetic analysis of AMPA and Kainate classes in protostomes was inferred using 110 sequences from 15 protostome species (*Figure 3—source data 2*) and 37 sequences from deuterostomes, of which 4 GluN1 proteins were used as an outgroup. The mGluR tree was constructed with 149 proteins from 29 non-vertebrate species, 38 mGluRs from vertebrate species and 10 sequences from vertebrate metabotropic GABA receptors, used as an outgroup (*Figure 6—source data 2*).

Protein sequences were aligned with the MUSCLE algorithm (*Edgar, 2004*), included in the software package MEGA6 (*Tamura et al., 2013*) with default parameters. ProtTest v3.4.2 was used to establish the best evolutionary model (*Darriba et al., 2011*). Trees were constructed using MrBayes v3.2.6 (*Ronquist et al., 2012*) for Bayesian inference and IQ-TREE (*Nguyen et al., 2015*) for Maximum-likelihood analysis. For Bayesian inference phylogenies were stopped when standard deviation was below 0.01 and its value was fluctuating but not decreasing. Markov chain Monte Carlo (MCMC) was used to approximate the posterior probability of the Bayesian trees. Bayesian analyses included two independent MCMC runs, each using four parallel chains composed of three heated and one cold chain. Twenty-five % of initial trees were discarded as burn-in. Convergence was assessed when potential scale reduction factor (PSRF) value was between 1.002 and 1.000. In Maximum-likelihood analysis the starting tree was estimated using a neighbor-joining method and branch support was obtained after 1000 iterations of ultrafast bootstrapping (*Hoang et al., 2018*). Gene/protein names were given based on their position in the tree. Phylogenetic trees were rendered using FigTree (http://tree.bio.ed.ac.uk/software/figtree/). Phylogenetic calculations were performed at the IBB - UAB heterogeneous computer cluster 'Celler' and at the CIPRES science gateway (RRID: SCR_008439) (*Miller et al., 2010*).

## Collection and housing of animals

*Branchiostoma lanceolatum* adults were collected in the bay of Argelès-sur-Mer, France (latitude 42° 32′ 53′ N and longitude 3° 03′ 27′ E) with a specific permission delivered by the Prefect of Region Provence Alpes Côte d'Azur. *B. lanceolatum* is not a protected species. Animals were kept in tanks with seawater at 17°C under natural photoperiod.

## RNA isolation, cDNA synthesis and quantitative gene expression (qPCR)

Adult amphioxus (*B. lanceolatum*) were anesthetized in 0.1% diethyl pyrocarbonate (DEPC; Sigma, D5758) PBS buffer. Animals were sacrificed by cutting the most anterior part of the body. The nerve chord was surgically extracted from the animal while submerged in DEPC-PBS using a magnifying glass. Individual nerve chords were snap frozen in liquid nitrogen and stored at −80°C until use. RNA was extracted from whole animals or from dissected nerve chords. Ten nerve chords were used for each RNA extraction, so that biological variability between individuals could be normalized. The tissue was homogenized in 1 mL of TRI Reagent (Sigma, T9424) using a Polytron homogenizer. Homogenates were transferred into an Eppendorf tube and incubated 5 min at room temperature

(RT) before adding 100 μL of 1-bromo-3-cloropropane. Tubes were vigorously mixed by vortexing for 10–15 s, incubated 15 min at RT and centrifuged at 13000 rpm for 15 min at 4°C. RNA was precipitated from the aqueous phase with 500 μL of isopropanol and 20 μg of glycogen. Tubes were frozen for 1 hr at −80°C and then thawed, incubated at RT for 10 min and centrifuged at 13000 rpm for 10 min at 4°C. The RNA pellet was washed twice with 500 μL of 75% ethanol and air-dried. cDNA was synthesized from 0.5 μg of total RNA. One μL of Oligo(dT)15 (Promega), 1 μL of 10 mM dNTP mix (Biotools), RNA and DEPC distilled water were mixed in a PCR tube to a final volume of 14 μL. This mix was incubated at 65°C for 5 min in a T100 Thermal Cycler (BioRad). After cooling tubes on ice for 1 min, we added 4 μL of First Strand 5x buffer, 1 μL of 0.1 M DTT and 1 μL of Super-Script III (Invitrogen). Tubes were placed in a T100 Thermal Cycler (BioRad) with the following program: 60 min at 50°C, 15 min at 70°C. RNA expression levels were determined using qPCR and the GAPDH gene used as a reference. Primers used for qPCR analysis of iGluRs are in *Figure 4—figure supplement 2* and those used for mGluR qPCR in *Figure 6—figure supplement 4*. qPCR data for iGluRs and mGluRs are given in *Figure 4—source data 1* and *Figure 6—source data 4*, respectively.

cDNA from nerve chord and whole body samples was diluted 1:10 for the glutamate receptor gene reactions, and 1:100 for the reference gene reaction. For each gene 2.5 μL of diluted cDNA were added to 5 μL of iTaq Universal SYBR Green Supermix (Bio-Rad), along with 0.5 μL of each primer and 1.5 μL of RNase free water. qPCR was run in a C1000 Touch thermocycler combined with the optic module CFX96. Three technical replicates were performed for all genes analyzed. Primer pairs were designed to detect the expression levels of each glutamate receptor (*Figure 4—figure supplement 2* and *Figure 6—figure supplement 4*). *B. belcheri* glutamate receptor sequences were aligned with the genomic sequence of *B. lanceolatum*, and high identity fragments were used to design primers. All primers were 20–25 base pair long, had GC content over 40–45% and a Tm between 60–65°C. Primers were designed to obtain amplicons between 140–270 base pairs. Values of normalized expression were statistically analyzed using GraphPad Prism5. No outliers were identified and no data points were excluded. Comparisons between whole body and nerve chord expression levels were done with Student's T-Test for unpaired samples or the Welch variant of the Student's T-Test for samples with different variance. For multiple comparisons between the expression levels of genes belonging to the same class one-way ANOVA analysis was performed using Tukey's Post-Hoc test.

### *Grie1* and *Grie7* gene synthesis

*Grie1* and *Grie7* genes were selected for transient expression in the mammalian cell line HEK293T. We prepared two constructs for each gene. We first introduced an immuno-tag in the N-terminus before the first element of secondary structure. For *Grie1* we used the c-Myc tag, which was placed after residue 39, and for *Grie7* we used the hemagglutinin (HA) tag introduced after residue 10 of the wild-type sequence. The second set of constructs prepared substituted the wild type N-terminal sequence for the signal peptide from rat GluA2 while maintaining the immuno-tags (*Figure 4—figure supplement 1*). Codon-optimized genes for expression in human cells were synthesized and cloned into pICherryNeo (Addgene, 52119) and pIRES2_EGFP (Addgene 6029–1) by the Invitrogen GeneArt Gene Synthesis service.

### Cell line

All expression experiments were done with a mycoplasma-free HEK293T cell line kindly provided by Prof. F. Ciruela (Universitat de Barcelona) and purchased from the American Type Culture Collection (ATCC, CRL-3216, RRID: CVCL_0063). The ATCC has confirmed the identity of HEK293T by STR profiling (STR Profile; CSF1PO: 11,12; D13S317: 12,14; D16S539: 9,13; D5S818: 8,9; D7S820: 11; TH01: 7, 9.3; TPOX: 11; vWA: 16,19; Amelogenin: X). After the purchase of the cell line, mycoplasma tests are performed in the laboratory on every new defrosted aliquot. The kit used for mycoplasma detection is PlasmoTest (Invivogen, code: rep-pt1).

## Expression of GluE1 and GluE7 in HEK293T cells and analysis of plasma membrane trafficking

HEK293T cells were maintained in Dulbecco's Modified Eagle Medium (DMEM) supplemented with 10% FBS and 1% Antibotic-Antimycotic (Gibco) in a humidified incubator at 5% $CO_2$ air and 37°C. The day before transfection, cells were plated onto poly-D-lysine coated coverslips in 6-well plates, to reach 60–80% confluence. HEK293T cells were transiently transfected with the following plasmids: empty pIRES2-EGFP, pIRES2-EGFP containing the Grie7_Bbe gene, empty pICherryNeo and pICherryNeo containing Grie1_Bla. Cells were transfected using 3 µg of polyethylenimine and 1 µg of plasmid DNA for each ml of non-supplemented DMEM. Cells were incubated 4–5 hr with transfection medium without supplementation, which was then removed and replaced by supplemented medium. Twenty-four hours after transfection the medium was removed and cells were washed 3 times with PBS. For surface receptor staining, cells were blocked in 2% BSA in PBS for 10 min at 37°C, and incubated for 25 min at 37°C with primary antibodies against HA (Covance, MMS-101P, RRID: AB_291259), c-Myc (Cell Signalling, 2272S, RRID: AB_10692100) or GluA2 (Millipore, MAB397, RRID: AB_2113875). HA and GluA2 antibodies were diluted 1:200 and c-Myc 1:100 in DMEM without supplementation. Cells were washed 3 times with PBS, fixed in 4% paraformaldehyde (PFA) for 15 min at RT, rinsed in PBS and incubated 1 hr at 37°C with secondary antibodies Alexa Fluor 555 donkey anti-mouse IgG (H + L) (A-31570, Invitrogen, RRID: AB_2536180) and Alexa Fluor 647 goat anti-rabbit IgG (H + L) highly cross-adsorbed (Life Technologies, A-21245, RRID: AB_2535813), diluted 1:1000 and 1:500 in PBS, respectively. Finally, coverslips were washed and mounted onto slides with Fluoroshield with DAPI (Sigma-Aldrich, F6057). For intracellular labeling cells were first fixed in 4% PFA for 15 min at RT, permeabilized with 0.2% Triton X-100 in PBS for 10 min, and finally blocked with PBS containing 2% BSA and 0.2% Triton X-100 for 20 min. Primary antibodies against HA (Covance, MMS-101P, RRID: AB_291259) and GluA2 (Millipore, MAB397, RRID: AB_2113875) were diluted 1:1000 and c-Myc (Cell Signalling, 2272S, RRID: AB_10692100) antibody was prepared at 1:100 in PBS. Incubation lasted 25 min at 37°C. Secondary antibody incubations and coverslip mounting were done in the same way as for non-permeabilized cells. Cells were examined using a confocal laser-scanning microscope (Zeiss LSM 700) with a 63x oil objective.

## Western blot and native gel electrophoresis

HEK293T cells were grown in 6-well plates as described previously and transfected with plasmids expressing amphioxus GluE1, GluE7 or GluA2. Twenty-four hours after transfection cells were rinsed with PBS and the content of 4 wells was resuspended in solubilization buffer (PBS containing 2% N-dodecyl-α-maltopyranoside (DDM; D310HA, Anatrace) and the protease inhibitors mix cOmplete EDTA-free Protease Inhibitor Cocktail, Roche). Cell lysates were homogenized in a Dounce homogenizer in ice with 20 strokes and kept under orbital agitation for 1 hr at 4°C. Lysates were centrifuged at 89000xg in a Beckman TLA120.2 rotor for 40 min at 4°C. The supernatant containing solubilized membrane proteins was recovered in a new tube and stored at −20°C until used.

For native gel electrophoresis proteins were resolved in a Mini-PROTEAN TGX Gel 4–20% (Bio-Rad). Samples were mixed with Native Sample Buffer (Bio-Rad) and run along with HiMark Pre-Stained Protein Standard (Life Technologies). Electrophoresis was performed in ice at a constant voltage of 100 V for 180 min. Gels were transferred at constant current (35 mA) to polyvinylidene fluoride (PVDF) membranes overnight (16–18 hr) at 4°C. After transfer, membranes were blocked for 1 hr with Odyssey Blocking Buffer (Li-cor) in TBS, and incubated overnight at 4°C with primary antibodies anti-HA (Covance, MMS-101P, RRID: AB_291259), anti-c-Myc (Cell Signaling, 2272S, RRID: AB_10692100) or anti-GluA2 (Millipore, MAB397, RRID: AB_2113875) diluted 1:1000 in TTBS (TBS containing 0.05% Tween-20). After three 15 min washes in TTBS, membranes were incubated with donkey anti-mouse (Li-cor, 926–32212, RRID: AB_621847) and donkey anti-rabbit (Li-cor, 926–68073, RRID: AB_10954442) diluted 1:7500 in TTBS for 1 hr. Blots were analyzed in an Odyssey scanner (Li-cor).

For denaturing gel electrophoresis (SDS-PAGE) protein lysates were denatured by adding loading sample buffer 10x (500 mM Tris-HCl pH 7.4, 20% SDS, 10% β-mercaptoethanol, 10% glycerol and 0.04% bromophenol blue), and incubated for 5 min at 95°C. Protein lysates were loaded in a 10% SDS- polyacrylamide gel and separated at a constant current (25 mA). Gels were transferred at a constant voltage of 100 V for 90 min in ice. Membranes were blocked for 1 hr with Odyssey Blocking

Buffer in TBS, and incubated overnight at 4°C with the same primary antibodies at the same dilution as for native gels in TBS containing 0.1% Tween 20. After three 15 min washes in TTBS, membranes were incubated with secondary antibodies as above. Blots were analyzed in an Odyssey scanner.

## 3D modeling of GluE1 and GluE7

Models for full-length GluE1 and GluE7 were generated with RaptorX (*Källberg et al., 2012*) based on deposited three-dimensional crystal structures of the full-length AMPA-subtype ionotropic glutamate receptor from *Rattus norvegicus*, GluA2, bound to competitive antagonists (PDB codes 4U4G (*Yelshanskaya et al., 2014*) and 3KG2 (*Sobolevsky et al., 2009*), respectively). Models of their respective ligand binding domains were generated with SWISS-MODEL (*Biasini et al., 2014*) using the atomic-resolution crystal structure of the rat GluA2 LBD bound to glutamate as template (PDB code 4YU0). Model quality was assessed with MolProbity (http://molprobity.biochem.duke.edu/, RRID: SCR_014226). MolProbity scores for all models are given in *Figure 1—source data 4*. Models were inspected with MIFit (*Smith, 2010*) and figures were prepared with PyMOL (www.pymol.org).

## Electrophysiology

Cells were visualized with an inverted epifluorescence microscope (AxioVert A.1, Zeiss) and were constantly perfused at 22–25°C with an extracellular solution containing (in mM): 145 NaCl, 2.5 KCl, 2 $CaCl_2$, 1 $MgCl_2$, 10 HEPES and 10 glucose (pH = 7.42 with NaOH; 305 mOsm/Kg). Microelectrodes were filled with an intracellular solution containing (in mM): 145 CsCl, 2.5 NaCl, 1 Cs-EGTA, 4 MgATP, 10 HEPES (pH = 7.2 with CsOH; 295 mOsm/Kg). Electrodes were fabricated from borosilicate glass (1.5 mm o.d., 1.16 i.d., Harvard Apparatus) pulled with a P-97 horizontal puller (Sutter Instruments) and polished with a forge (MF-830, Narishige) to a final resistance of 2–4 MΩ. Currents were recorded with an Axopatch 200B amplifier filtered at 1 KHz and digitized at 5 KHz using Digidata 1440A interface with pClamp 10 software (Molecular Devices Corporation).

Whole-cell macroscopic currents were recorded from isolated or coupled pairs of mCherry or EGFP positive HEK293T cells. Rapid application (<1 ms exchange) of agonists (500 ms pulses) at a membrane potential of −60 mV was achieved by means of a theta-barrel tool (1.5 mm o.d.; Sutter Instruments) coupled to a piezoelectric translator (P-601.30; Physik Instrumente). One barrel contained extracellular solution diluted to 96% with $H_2O$ and the other barrel contained 10 mM of the amino acid solution. For measuring current-voltage relationships, 500 ms agonist jumps were applied at different membrane voltages (−80 mV to +80 mV in 20 mV steps) and peak currents were fitted to a $5^{th}$ order polynomial function. To study recovery from desensitization, a two-pulse protocol (500 ms each) was used in which a first pulse was applied followed by a second pulse at different time intervals (from 2.5 s to 25 s). The paired pulses were separated 30–60 s to allow full recovery from desensitization. To estimate the percentage of recovery, the magnitude of peak current at the second pulse (P2) was compared with the first one (P1). Electrophysiological recordings were analyzed using IGOR Pro (Wavemetrics Inc.) with NeuroMatic (Jason Rothman, UCL, RRID: SCR_004186).

## Acknowledgments

We thank I Gich (Biomedical Research Institute Sant Pau) for support in biostatistics and Dr. Xavier Daura and Oscar Conchillo (Computational Biology Group and Data Center from the Institut de Biotecnologia i Biomedicina (UAB). DRV, GG, RRV, JL, PFP, and AB work at a research institute supported by the CERCA Programme/Generalitat de Catalunya.

## Additional information

### Funding

| Funder | Grant reference number | Author |
| --- | --- | --- |
| Ministerio de Economía y Competitividad | BFU2012-34398 | David Ramos-Vicente<br>Gemma Gou<br>Rita Reig-Viader<br>Javier Luís<br>Àlex Bayés |

| | | |
|---|---|---|
| Ministerio de Economía y Competitividad | BFU2015-69717-P | David Ramos-Vicente<br>Gemma Gou<br>Rita Reig-Viader<br>Javier Luís<br>Àlex Bayés |
| Seventh Framework Programme | 304111 | David Ramos-Vicente<br>Gemma Gou<br>Rita Reig-Viader<br>Javier Luís<br>Àlex Bayés |
| Ministerio de Economía y Competitividad | RYC-2011-08391 | Àlex Bayés |
| Ministerio de Economía y Competitividad | RYC-2010-06210 | Nerea Roher |
| China Scholarship Council | CSC-2013-06300075 | Jie Ji |
| Ministerio de Economía y Competitividad | SAF2014-57994-R | Pablo Fuentes-Prior |
| Ministerio de Economía y Competitividad | AGL2015-65129-R | Jie Ji<br>Nerea Roher |
| Generalitat de Catalunya | SGR-345-2014 | David Ramos-Vicente<br>Gemma Gou<br>Rita Reig-Viader<br>Javier Luís<br>Àlex Bayés |
| Ministerio de Economía y Competitividad | BFU2014-57562-P | David Soto |
| Centre National de la Recherche Scientifique | ANR-16-CE12-0008-01 | Hector Escriva |
| Ministerio de Economía y Competitividad | BFU2017-83317-P | David Soto |
| Ministerio de Economía y Competitividad | RD16/0008/0014 | David Soto |

The funders had no role in study design, data collection and interpretation, or the decision to submit the work for publication.

### Author contributions

David Ramos-Vicente, Formal analysis, Validation, Investigation, Methodology, Writing—review and editing; Jie Ji, David Soto, Investigation, Methodology, Writing—review and editing; Esther Gratacòs-Batlle, Supervision, Investigation, Writing—review and editing; Gemma Gou, Rita Reig-Viader, Investigation, Writing—review and editing; Javier Luís, Pablo Fuentes-Prior, Formal analysis, Investigation, Methodology, Writing—review and editing; Demian Burguera, Enrique Navas-Perez, Methodology, Writing—review and editing; Jordi García-Fernández, Supervision, Methodology, Writing—review and editing; Hector Escriva, Conceptualization, Supervision, Investigation, Writing—review and editing; Nerea Roher, Funding acquisition, Investigation, Methodology, Writing—review and editing; Àlex Bayés, Conceptualization, Formal analysis, Supervision, Funding acquisition, Methodology, Writing—original draft, Project administration, Writing—review and editing

### Author ORCIDs

David Ramos-Vicente (iD) http://orcid.org/0000-0002-2730-0850
David Soto (iD) http://orcid.org/0000-0001-7995-3805
Àlex Bayés (iD) http://orcid.org/0000-0002-5265-6306

### Decision letter and Author response

Decision letter https://doi.org/10.7554/eLife.35774.119
Author response https://doi.org/10.7554/eLife.35774.120

## Additional files

**Supplementary files**

• Transparent reporting form

DOI: https://doi.org/10.7554/eLife.35774.032

### Data availability

All data generated or analysed during this study are included in the manuscript and supporting files. Source data files have been provided for Figures 1, Figure 1 - figure supplement 1, Figure 1 - figure supplement 3, Figure 1 - figure supplement 4, Figure2, Figure 4, Figure 4 - figure supplement 1 and Figure 4 - figure supplement 3. Genes from the sponges *Oscarella carmela*, *Sycon cilliatum* and *Leucosolenia complicata* were obtained from Compagen (http://www.compagen.org/index.html).

The following previously published datasets were used:

| Author(s) | Year | Dataset title | Dataset URL | Database and Identifier |
|---|---|---|---|---|
| Sea Urchin Genome Sequencing Consortium, Sodergren E, Weinstock GM, Davidson EH, Cameron RA, Gibbs RA, Angerer RC, Angerer LM, Arnone MI, Burgess DR, Burke RD, Coffman JA, Dean M, Elphick MR, Ettensohn CA, Foltz KR, Hamdoun A, Hynes RO, Klein WH, Marzluff W, McClay DR, Morris RL, Mushegian A, Rast JP, Smith LC, Thorndyke MC, Vacquier VD, Wessel GM, Wray G, Zhang L, Elsik CG, Ermolaeva O, Hlavina W, Hofmann G, Kitts P, Landrum MJ, Mackey AJ, Maglott D, Panopoulou G, Poustka AJ, Pruitt K, Sapojnikov V, Song X, Souvorov A, Solovyev V, Wei Z, Whittaker CA, Worley K, Durbin KJ, Shen Y, Fedrigo O, Garfield D, Haygood R, Primus A, Satija R, Severson T, Gonzalez-Garay ML, Jackson AR, Milosavljevic A, Tong M, Killian CE, Livingston BT, Wilt FH, Adams N, Bellé R, Carbonneau S, Cheung R, Cormier P, Cosson B, Croce J, Fernandez-Guerra A, Geneviève AM, Goel M, Kelkar H, Morales J, Mulner- | 2006 | The genome of the sea urchin Strongylocentrotus purpuratus | https://metazoa.ensembl.org/Strongylocentrotus_purpuratus/Info/Index | Ensembl Metazoa, Strongylocentrotus purpuratus |

Lorillon O, Robertson AJ, Goldstone JV, Cole B, Epel D, Gold B, Hahn ME, Howard-Ashby M, Scally M, Stegeman JJ, Allgood EL, Cool J, Judkins KM, McCafferty SS, Musante AM, Obar RA, Rawson AP, Rossetti BJ, Gibbons IR, Hoffman MP, Leone A, Istrail S, Materna SC, Samanta MP, Stolc V, Tongprasit W, Tu Q, Bergeron KF, Brandhorst BP, Whittle J, Berney K, Bottjer DJ, Calestani C, Peterson K, Chow E, Yuan QA, Elhaik E, Graur D, Reese JT, Bosdet I, Heesun S, Marra MA, Schein J, Anderson MK, Brockton V, Buckley KM, Cohen AH, Fugmann SD, Hibino T, Loza-Coll M, Majeske AJ, Messier C, Nair SV, Pancer Z, Terwilliger DP, Agca C, Arboleda E, Chen N, Churcher AM, Hallböök F, Humphrey GW, Idris MM, Kiyama T, Liang S, Mellott D, Mu X, Murray G, Olinski RP, Raible F, Rowe M, Taylor JS, Tessmar-Raible K, Wang D, Wilson KH, Yaguchi S, Gaasterland T, Galindo BE, Gunaratne HJ, Juliano C, Kinukawa M, Moy GW, Neill AT, Nomura M, Raisch M, Reade A, Roux MM, Song JL, Su YH, Townley IK, Voronina E, Wong JL, Amore G, Branno M, Brown ER, Cavalieri V, Duboc V, Duloquin L, Flytzanis C, Gache C, Lapraz F, Lepage T, Locascio A, Martinez P, Matassi G, Matranga V, Range R, Rizzo F, Röttinger E, Beane W, Bradham C, Byrum C, Glenn T, Hussain S, Manning G, Miranda E, Thomason R, Walton K,

| | | | | |
|---|---|---|---|---|
| Wikramanayke A, Wu SY, Xu R, Brown CT, Chen L, Gray RF, Lee PY, Nam J, Oliveri P, Smith J, Muzny D, Bell S, Chacko J, Cree A, Curry S, Davis C, Dinh H, Dugan-Rocha S, Fowler J, Gill R, Hamilton C, Hernandez J, Hines S, Hume J, Jackson L, Jolivet A, Kovar C, Lee S, Lewis L, Miner G, Morgan M, Nazareth LV, Okwuonu G, Parker D, Pu LL, Thorn R, Wright R | | | | |
| Simakov O, Marletaz F, Cho S-J, Edsinger-Gonzales E, Havlak P, Hellsten U, Kuo DH, Larsson T, Lv J, Arendt D, Savage R, Osoegawa K, de Jong P, Grimwood J, Chapman JA, Shapiro H, Aerts A, Otillar RP, Terry AY, Boore JL, Grigoriev IV, Lindberg DR, Seaver EC, Weisblat DA, Putnam NH | 2013 | Insights into bilaterian evolution from three spiralian genomes | https://metazoa.ensembl.org/Lottia_gigantea/Info/Index | Ensembl Metazoa, Lottia gigantea |
| Chipman AD, Ferrier DE, Brena C, Qu J, Hughes DS, Schröder R, Torres-Oliva M, Znassi N, Jiang H, Almeida FC, Alonso CR, Apostolou Z, Aqrawi P, Arthur W, Barna JC, Blankenburg KP, Brites D, Capella-Gutiérrez S, Coyle M, Dearden PK, Du Pasquier L, Duncan EJ, Ebert D, Eibner C, Erikson G, Evans PD, Extavour CG, Francisco L, Gabaldón T, Gillis WJ, Goodwin-Horn EA, Green JE, Griffiths-Jones S, Grimmelikhuijzen CJ, Gubbala S, Guigó R, Han Y, Hauser F, Havlak P, Hayden L, Helbing S, Holder M, Hui JH, Hunn JP, Hunnekuhl VS, Jackson L, Javaid M, Jhangiani SN, Jiggins FM, Jones TE, Kaiser TS, Kalra D, Kenny NJ, Kor- | 2014 | The first myriapod genome sequence reveals conservative arthropod gene content and genome organisation in the centipede Strigamia maritima | https://metazoa.ensembl.org/Strigamia_maritima/Info/Index?db=core | Ensembl Metazoa, Strigamia maritima |

| | | | | |
|---|---|---|---|---|
| china V, Kovar CL, Kraus FB, Lapraz F, Lee SL, Lv J, Mandapat C, Manning G, Mariotti M, Mata R, Mathew T, Neumann T, Newsham I, Ngo DN, Ninova M, Okwuonu G, Ongeri F, Palmer WJ, Patil S, Patraquim P, Pham C, Pu LL, Putman NH, Rabouille C, Ramos OM, Rhodes AC, Robertson HE, Robertson HM, Ronshaugen M, Rozas J, Saada N, Sánchez-Gracia A, Scherer SE, Schurko AM, Siggens KW, Simmons D, Stief A, Stolle E, Telford MJ, Tessmar-Raible K, Thornton R, van der Zee M, von Haeseler A, Williams JM, Willis JH, Wu Y, Zou X, Lawson D, Muzny DM, Worley KC, Gibbs RA, Akam M | | | | |
| Simakov O, Marletaz F, Cho S-J, Edsinger-Gonzales E, Havlak P, Hellsten U, Kuo DH, Larsson T, Lv J, Arendt D, Savage R, Osoegawa K, de Jong P, Grimwood J, Chapman JA, Shapiro H, Aerts A, Otillar RP, Terry AY, Boore JL, Grigoriev IV, Lindberg DR, Seaver EC, Weisblat DA, Putnam NH | 2013 | Insights into bilaterian evolution from three spiralian genomes | https://metazoa.ensembl.org/Capitella_teleta/Info/Index | Ensembl Metazoa, Capitella teleta |
| Elsik CG, Worley KC, Bennett AK, Beye M, Camara F, Childers CP, de Graaf DC, Debyser G, Deng J, Devreese B, Elhaik E, Evans JD, Foster LJ, Graur D, Guigo R; HGSC production teams, Hoff KJ, Holder ME, Hudson ME, Hunt GJ, Jiang H, Joshi V, Khetani RS, Kosarev P, Kovar CL, Ma J, Maleszka R, Moritz RF, Munoz-Torres MC, Murphy TD, Muzny DM, Newsham IF, Reese JT, Robertson HM, Robinson GE, Rueppell O, | 2014 | Finding the missing honey bee genes: lessons learned from a genome upgrade | https://metazoa.ensembl.org/Apis_mellifera/Info/Index | Ensembl Metazoa, Apis mellifera |

| | | | | |
|---|---|---|---|---|
| Solovyev V, Stanke M, Stolle E, Tsuruda JM, Vaarenbergh MV, Waterhouse RM, Weaver DB, Whitfield CW, Wu Y, Zdobnov EM, Zhang L, Zhu D | | | | |
| Putnam NH, Srivastava M, Hellsten U, Dirks B, Chapman J, Salamov A, Terry A, Shapiro H, Lindquist E, Kapitonov VV, Jurka J, Genikhovich G, Grigoriev IV, Lucas SM, Steele RE, Finnerty JR, Technau U, Martindale MQ | 2007 | Sea anemone genome reveals ancestral eumetazoan gene repertoire and genomic organization | https://metazoa.ensembl.org/Nematostella_vectensis/Info/Index | Ensembl Metazoa, Nematostella vectensis |
| Srivastava M, Begovic E, Chapman J, Putnam NH, Hellsten U, Kawashima T, Kuo A, Mitros T, Salamov A, Carpenter ML, Signorovitch AY, Moreno MA, Kamm K, Grimwood J, Schmutz J, Shapiro H, Grigoriev IV, Buss LW, Schierwater B, Dellaporta SL | 2008 | The Trichoplax genome and the nature of placozoans | https://metazoa.ensembl.org/Trichoplax_adhaerens/Info/Index?db=core | Ensembl Metazoa, Trichoplax adhaerens |
| Ryan JF, Pang K, Schnitzler CE, Nguyen AD, Moreland RT, Simmons DK, Koch BJ, Francis WR, Havlak P; NISC Comparative Sequencing Program, Smith SA, Putnam NH, Haddock SH, Dunn CW, Wolfsberg TG, Mullikin JC, Martindale MQ | 2013 | The genome of the ctenophore Mnemiopsis leidyi and its implications for cell type evolution | https://metazoa.ensembl.org/Mnemiopsis_leidyi/Info/Index | Ensembl Metazoa, Mnemiopsis leidyi |
| Arabidopsis Genome Initiative | 2000 | Analysis of the genome sequence of the flowering plant Arabidopsis thaliana | https://www.arabidopsis.org/browse/genefamily/Ionchannels.jsp | The Arabidopsis Information Resource, TAIR |
| Genome Reference Consortium | 2017 | Genome Reference Consortium Human Build 38 patch release 12 (GRCh38.p12) | https://www.ensembl.org/Homo_sapiens/Info/Index?db=core | Ensembl, Homo_sapiens |
| Genome Reference Consortium | 2017 | Genome Reference Consortium Mouse Build 38 patch release 6 (GRCm38.p6) | https://www.ensembl.org/Mus_musculus/Info/Index | Ensembl, Mus_musculus |
| Genome Reference Consortium | 2017 | Genome Reference Consortium Zebrafish Build 11 | https://www.ensembl.org/Danio_rerio/Info/Index | Ensembl, danio_rerio |
| International Chicken Genome Consortium | 2015 | Gallus_gallus-5.0 | https://www.ensembl.org/Gallus_gallus/Info/Index?db=core | Ensembl, Gallus_gallus |
| DOE Joint Genome Institute | 2009 | v4.2 | https://www.ensembl.org/Xenopus_tropicalis/Info/Index | Ensembl, Xenopus 4.2 |

| | | | | |
|---|---|---|---|---|
| Simakov O, Kawashima T, Marlétaz F, Jenkins J, Koyanagi R, Mitros T, Hisata K, Bredeson J, Shoguchi E, Gyoja F, Yue JX, Chen YC, Freeman RM Jr, Sasaki A, Hikosaka-Katayama T, Sato A, Fujie M, Baughman KW, Levine J, Gonzalez P, Cameron C, Fritzenwanker JH, Pani AM, Goto H, Kanda M, Arakaki N, Yamasaki S, Qu J, Cree A, Ding Y, Dinh HH, Dugan S, Holder M, Jhangiani SN, Kovar CL, Lee SL, Lewis LR, Morton D, Nazareth LV, Okwuonu G, Santibanez J, Chen R, Richards S, Muzny DM, Gillis A, Peshkin L, Wu M, Humphreys T, Su YH, Putnam NH, Schmutz J, Fujiyama A, Yu JK, Tagawa K, Worley KC, Gibbs RA, Kirschner MW, Lowe CJ, Satoh N, Rokhsar DS | 2015 | Hemichordate genomes and deuterostome origins | https://metazome.jgi. doe.gov/pz/portal.html#! search?show=KEY-WORD&method=Org_ Skowalevskii_er | Metazome, v3.2 |
| Huang S, Chen Z, Yan X, Yu T, Huang G, Yan Q, Pontarotti PA, Zhao H, Li J, Yang P, Wang R, Li R, Tao X, Deng T, Wang Y, Li G, Zhang Q, Zhou S, You L, Yuan S, Fu Y, Wu F, Dong M, Chen S | 2014 | Decelerated genome evolution in modern vertebrates revealed by analysis of multiple lancelet genomes | http://genome.bucm. edu.cn/lancelet/search. php | LanceletDB, B. belcheri_HapV2(v7h2) _cds |
| Satou Y, Kawashima T, Shoguchi E, Nakayama A | 2005 | An integrated database of the ascidian, Ciona intestinalis: towards functional genomics | https://www.ncbi.nlm. nih.gov/genome/49 | NCBI Genome, 49 |
| Martín-Durán JM, Hejnol A | 2015 | The study of Priapulus caudatus reveals conserved molecular patterning underlying different gut morphogenesis in the Ecdysozoa | https://www.ncbi.nlm. nih.gov/genome/?term= Priapulus+caudatus | NCBI Genome, 789 |
| Hall MR, Kocot KM, Baughman KW, Fernandez-Valverde SL, Gauthier MEA, Hatleberg WL, Krishnan A, McDougall C, Motti CA, Shoguchi E, Wang T, Xiang X, Zhao M, Bose U, Shinzato C, Hisata K, Fujie M, Kanda M, Cummins SF, Satoh N, Degnan SM, Degnan BM. | 2017 | The crown-of-thorns starfish genome as a guide for biocontrol of this coral reef pest | https://www.ncbi.nlm. nih.gov/genome/?term= acanthaster+planci | NCBI Genome, 7870 |

| | | | | |
|---|---|---|---|---|
| Simakov O, Takeshi Kawashima, Ferdinand Marletaz, Jerry Jenkins, Ryo Koyanagi, Therese Mitros, Kanako Hisata, Jessen Bredeson, Eiichi Shoguchi, Fuki Gyoja, Jia-Xing Yue, Robert Freeman, Akane Sasaki, Tomoe Hikosaka-Katayama, Atsuko Sato, Manabu Fujie, Kenneth W. Baughman, Judith Levine, Paul Gonzalez, Christopher Cameron, Jens Fritzenwanker, Ariel Pani, Hiroki Goto, Miyuki Kanda, Nana Arakaki, Shinichi Yamasaki, Jiaxin Qu, Andrew Cree, Yan Ding, Huyen H. Dinh, Shannon Dugan, Michael Holder, Shalini N. Jhangiani, Christie L. Kovar, Sandra L. Lee, Lora R. Lewis, Donna Morton, Lynne V. Nazareth, Geoffrey Okwuonu, Jireh Santibanez, Rui Chen, Stephen Richards, Donna M. Muzny, Leonid Peshkin, Michael Wu, Tom Humphreys, Yi-Hsien Su, Nicholas Putnam, Jeremy Schmutz, Asao Fujiyama, Jr-Kai Yu, Kunifumi Tagawa, Kim C Worley, Richard A. Gibbs, Marc W. Kirschner, Christopher J Lowe, Noriyuki Satoh, Daniel S Rokhsar, John Gerhart | 2015 | Hemichordate genomes and deuterostome origins | http://marinegenomics. oist.jp/acornworm/viewer/info?project_id=33 | Marine Genomics Unit, Ptychodera flava |
| Shinzato C, Shoguchi E, Kawashima T, Hamada M, Hisata K, Tanaka M, Fujie M, Fujiwara M, Koyanagi R, Ikuta T, Fujiyama A, Miller DJ | 2011 | Using the Acropora digitifera genome to understand coral responses to environmental change | http://marinegenomics. oist.jp/coral/viewer/info? project_id=3 | Marine Genomics Unit, Acropora digitifera |
| Nicholas H. Putnam, Thomas Butts, David E. K. Ferrier, Rebecca F. Furlong, Uffe Hellsten, Takeshi Kawashima, Marc Robinson-Rechavi, Eiichi Shoguchi, Astrid Terry, Jr-Kai Yu, E'lia Benito- | 2008 | The amphioxus genome and the evolution of the chordate karyotype | https://genome.jgi.doe. gov/Brafl1/Brafl1.home. html | Joint Genome Institute, Brafl1 |

| Gutiérrez, Inna Dubchak, Jordi Garcia-Fernàndez, Jeremy J. Gibson-Brown, Igor V. Grigoriev, Amy C. Horton, Pieter J. de Jong, Jerzy Jurka, Vladimir V. Kapitonov, Yuji Kohara, Yoko Kuroki, Erika Lindquist, Susan Lucas, Kazutoyo Osoegawa, Len A. Pennacchio, Asaf A. Salamov, Yutaka Satou, Tatjana Sauka-Spengler, Jeremy Schmutz, Tadasu Shin-I, Atsushi Toyoda, Marianne Bronner-Fraser, Asao Fujiyama, Linda Z. Holland, Peter W. H. Holland, Nori Satoh, Daniel S. Rokhsar | | | | |
|---|---|---|---|---|
| Kim S, Martin KC | 2015 | Neuron-wide RNA transport combines with netrin-mediated local translation to spatially regulate the synaptic proteome | https://www.ncbi.nlm.nih.gov/genome/443 | NCBI Genome, 443 |
| Zhang G, Fang X, Guo X, Li L, Luo R, Xu F, Yang P, Zhang L, Wang X, Qi H, Xiong Z, Que H, Xie Y, Holland PW, Paps J, Zhu Y, Wu F, Chen Y, Wang J, Peng C, Meng J, Yang L, Liu J, Wen B, Zhang N, Huang Z, Zhu Q, Feng Y, Mount A, Hedgecock D, Xu Z, Liu Y, Domazet-Lošo T, Du Y, Sun X, Zhang S, Liu B, Cheng P, Jiang X, Li J, Fan D, Wang W, Fu W, Wang T, Wang B, Zhang J, Peng Z, Li Y, Li N, Chen M, He Y, Tan F, Song X, Zheng Q, Huang R, Yang H, Du X, Chen L, Yang M, Gaffney PM, Wang S, Luo L, She Z, Ming Y, Huang W, Huang B, Zhang Y, Qu T, Ni P, Miao G, Wang Q, Steinberg CE, Wang H, Qian L, Zhang G, Liu X, Yin Y | 2012 | The oyster genome reveals stress adaptation and complexity of shell formation | https://www.ncbi.nlm.nih.gov/genome/10758 | NCBI Genome, 10 758 |
| Lim RS, Anand A, Nishimiya-Fujisawa C, Kobayashi S, Kai T | 2014 | Analysis of Hydra PIWI proteins and piRNAs uncover early evolutionary origins of the piRNA pathway | https://www.ncbi.nlm.nih.gov/genome/?term=Hydra+magnipapillata | NCBI Genome, 12836 |

| | | | | | |
|---|---|---|---|---|---|
| Prada C, Hanna B, Budd AF, Woodley CM, Schmutz J, Grimwood J, Iglesias-Prieto R, Pandolfi JM, Levitan D, Johnson KG, Knowlton N, Kitano H, DeGiorgio M, Medina M | 2016 | Empty Niches after Extinctions Increase Population Sizes of Modern Corals | https://www.ncbi.nlm.nih.gov/genome/?term=orbicella+faveola | NCBI Genome, 13173 | |
| Leonid L. Moroz, Kevin M. Kocot, Mathew R. Citarella, Sohn Dosung, Tigran P. Norekian, Inna S. Povolotskaya, Anastasia P. Grigorenko, Christopher Dailey, Eugene Berezikov, Katherine M. Buckley, Andrey Ptitsyn, Denis Reshetov, Krishanu Mukherjee, Tatiana P. Moroz, Yelena Bobkova, Fahong Yu, Vladimir V. Kapitonov, Jerzy Jurka, Yuri V. Bobkov, Joshua J. Swore, David O. Girardo, Alexander Fodor, Fedor Gusev, Rachel Sanford, Rebecca Bruders, Ellen Kittler, Claudia E. Mills, Jonathan P. Rast, Romain Derelle, Victor V. Solovyev, Fyodor A. Kondrashov, Billie J. Swalla, Jonathan V. Sweedler, Evgeny I. Rogaev, Kenneth M. Halanych, Andrea B. Kohn | 2014 | The ctenophore genome and the evolutionary origins of neural systems | https://neurobase.rc.ufl.edu/pleurobrachia | NeuroBase, pleurobrachia | |
| Baumgarten S, Simakov O, Esherick LY, Liew YJ, Lehnert EM, Michell CT, Li Y, Hambleton EA, Guse A, Oates ME, Gough J, Weis VM, Aranda M, Pringle JR, Voolstra CR | 2015 | The genome of Aiptasia, a sea anemone model for coral symbiosis | https://www.ncbi.nlm.nih.gov/genome/?term=exaiptasia | NCBI Genome, 40858 | |
| Christian R. Voolstra, Yong Li, Yi Jin Liew, Sebastian Baumgarten, Didier Zoccola, Jean-François Flot, Sylvie Tambutté, Denis Allemand, Manuel Aranda | 2017 | Comparative analysis of the genomes of Stylophora pistillata and Acropora digitifera provides evidence for extensive differences between species of corals | https://www.ncbi.nlm.nih.gov/genome/?term=Acropora+digitifera | NCBI Genome, 10529 | |
| Luo YJ, Takeuchi T, Koyanagi R, Yamada L, Kanda M, Khalturina M, Fujie M, Yamasaki S, | 2015 | The Lingula genome provides insights into brachiopod evolution and the origin of phosphate biomineralization | https://www.ncbi.nlm.nih.gov/genome/?term=Lingula+anatina | NCBI Genome, 38582 | |

| | | | | |
|---|---|---|---|---|
| Endo K, Satoh N | | | | |
| Simakov O, Marletaz F, Cho SJ, Edsinger-Gonzales E, Havlak P, Hellsten U, Kuo DH, Larsson T, Lv J, Arendt D, Savage R, Osoegawa K, de Jong P, Grimwood J, Chapman JA, Shapiro H, Aerts A, Otillar RP, Terry AY, Boore JL, Grigoriev IV, Lindberg DR, Seaver EC, Weisblat DA, Putnam NH, Rokhsar DS | 2013 | Insights into bilaterian evolution from three spiralian genomes | https://www.ncbi.nlm.nih.gov/genome/?term=Helobdella+robusta | NCBI Genome, 15112 |
| Flot JF | 2013 | Genomic evidence for ameiotic evolution in the bdelloid rotifer Adineta vaga | https://www.ncbi.nlm.nih.gov/genome/?term=Adineta+vaga | NCBI Genome, 17312 |
| Terrapon N | 2014 | Molecular traces of alternative social organization in a termite genome | https://www.ncbi.nlm.nih.gov/genome/?term=Zootermopsis+nevadensis | NCBI Genome, 17755 |
| Sanggaard KW | 2014 | Stegodyphus mimosarum genome | https://www.ncbi.nlm.nih.gov/genome/?term=Stegodyphus+mimosarrum | NCBI Genome, 12925 |
| Mesquita R | 2015 | Rhodnius prolixus genome | https://www.ncbi.nlm.nih.gov/genome/?term=Rhodnius+prolixus | NCBI Genome, 447 |
| Vicoso B, Bachtrog D | 2015 | Numerous transitions of sex chromosomes in Diptera | https://www.ncbi.nlm.nih.gov/genome/2619 | NCBI Genome, 2619 |
| McKenna DD | 2016 | Anoplophora glabripennis genome | https://www.ncbi.nlm.nih.gov/genome/?term=Anoplophora+glabripennnis | NCBI Genome, 14033 |
| Albertin CB, Simakov O, Mitros T, Wang ZY, Pungor JR, Edsinger-Gonzales E, Brenner S, Ragsdale CW, Rokhsar DS | 2015 | The octopus genome and the evolution of cephalopod neural and morphological novelties | https://metazoa.ensembl.org/Octopus_bimaculoides/Info/Index | Ensembl Metazoa, PRJNA270931 |

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
