## [Decision Letter]

Thank you for submitting your article "Evolution of Glutamate Receptors Reveals Three Unreported Classes and Divergent Phyla-Specific Adaptations" for consideration by *eLife*. Your article has been reviewed by three peer reviewers, including Leon D. Islas as the Reviewing Editor and Reviewer #1, and the evaluation has been overseen by Richard Aldrich as the Senior Editor. The following individuals involved in review of your submission have agreed to reveal their identity: Mark L Mayer (Reviewer #3).

The reviewers have discussed the reviews with one another and the Reviewing Editor has drafted this decision to help you prepare a revised submission. We hope you will be able to submit the revised version within two months.

Summary:

This manuscript presents an extensive phylogenetic analysis of ionotropic and metabotropic glutamate receptors. The authors find two new families of iGluRs a and one family of mGluRs. Based on these new families, they also propose a new evolutionary tree for both classes of receptors. This work also provides functional data on one of the newly found families, AKD1 and show that they can work as excitatory glutamate receptors and are expressed in the nervous system of amphioxus, an early vertebrate.

Essential revisions:

Three reviewers have seen your manuscript. Although there is enthusiasm for the new findings reported, the reviewers found that there are major deficiencies with the analysis that led to the proposal of the new families of ionotropic glutamate receptors. In particular, the fact that the proposed AK1 family only shows up in one of the genealogies seems troubling and might be an indication that the methods used to get the phylogenies might not be optimal. The consensus among reviewers is that the phylogenetic analysis should be repeated using more robust methods (see the reviews), in order to discard the possibility that the AK1 group might be an artifact. If this is the case, it is possible that the AKD1 group and the new metabotropic receptor family will still be present, which is a significant finding. The proposed changes should be carried out in order for the manuscript to be reconsidered.

1) If amino acid sequences were used for phylogeny reconstruction (after alignment of nucleotide sequences), then this needs to be clarified in the methods. However, if DNA sequences were used during phylogeny reconstruction, then both phylogenetic analyses presented in Figure 1 need to be repeated to make sure the proposed new classes of receptor are recovered.

2) For the expression experiments of AKD1 receptors, authors need to clarify if maybe these receptors do no not express functionally, since they seem to lack signal peptides.

3) The analysis done to claim loss of certain receptors in some phyla (e.g. Cnidarians) need to be redone and the interpretation needs to be in line with the results of the analysis, since as it stands, seems overreaching.

Reviewer #1

The paper by Ramos-Vicente and collaborators presents bioinformatics and functional data pertaining the discovery of new classes of both ionotropic and metabotropic glutamate receptors. The authors have incorporated these new families into a revised evolutionary tree of glutamate receptors. These findings are exciting and expand our knowledge of the evolution of these important membrane proteins.

In the Introduction the following phrase needs clarification: Thesespecies present a genomic structure, gene sequence and gene contentresembling that of their ancestors…. We do not know the genome of their ancestors and therefore this is an inference.

Structural models of two proteins from the new AKD1 class from amphioxus where constructed using known iGluR structures. Although this is at first glance reasonable, a less biased approach should be taken. Models should also be produced letting the software choose the appropriate structural template. Also, the MolProbity scores should be given as well as a discussion of the probability that the sequences adjust to the templates.

In the Discussion section, there is no correlation between the number of iGluR genes and the nervous system complexity of an organism. Is it possible that this might reflect the fact that in lower organisms glutamate receptors also have non-synaptic functions, as chemoreceptors?

Please include more information on the iGluRs found on *A. thaliana* and their relation to the newly proposed phylogeny of iGluRs.

In the figures that plot mean values, it is not specified what measure of variability is being used (i.e. standard error, variance, etc.).

Reviewer #2

Conceptual Issues

In the manuscript "Evolution of Glutamate Receptors Reveals Three Unreported Classes and Divergent Phyla-Specific Adaptations", the authors conduct a phylogenetic analysis of animal glutamate receptors (both ionotropic and metabotropic) and claim to have found 3 new phylogenetically-distinct classes (2 ionotropic and 1 metabotropic). They propose that the classification mechanism for glutamate receptors must be revised to account for these new gene families. Further, they provide data from structural modeling and functional expression to argue that ionotropic receptors in one of these classes, AKD1, are gated by glycine and not glutamate.

While the manuscript addresses and interesting subject worthy of a high profile publication, there are issues with the phylogenetic analysis that must be addressed or clarified to understand how robustly the claims for new classes of ionotropic glutamate receptors are supported.

1) There appears to be a major flaw in the methods for constructing the phylogeny, at least based on my interpretation of the Materials and methods section. The authors state that they aligned coding DNA sequences in MEGA6 with the "codon option" activated. The codon option will avoid introducing gaps into codons, but it does not correct for systematic biases that can get introduced into protein phylogenies that cover a wide range of divergent species with distinct codon usage biases. For this reason, it is generally accepted that such phylogenies need to be constructed with direct use of the amino acid sequences, as stated in the documentation for MEGA6. If amino acid sequences were used for phylogeny reconstruction (after alignment of nucleotide sequences), then this needs to be clarified in the methods. However, if DNA sequences were used during phylogeny reconstruction, then both phylogenetic analyses presented in Figure 1 need to be repeated to make sure the proposed new classes of receptor are recovered.

2) There are major issues with congruency between the phylogenies presented in Figure 1 with and without the ctenophore glutamate receptors, and this directly impacts the proposed new groups that the authors identify. Specifically, the AK and AKD2 groups in the main figure are split do not hold up in the analysis with the ctenophores included. While AKD2 is not a firmly-proposed new class, AK is, and this is concerning. It certainly suggests that proposing the AK class is an over-reach of the data.

3) The two phylogenies do not use the same names for the majority of the same genes. This needs to be corrected so that the position of a gene within the two phylogenies can be readily discerned. For instance, in the proposed AKD1 group, two genes in the phylogeny with the ctenophores simply contain XP numbers.

4) Posterior probabilities calculated in Bayesian phylogeny reconstructions are not equivalent to Bootstrap support as the authors suggest – this should be corrected. Furthermore, posterior probabilities tend to be much higher than one would get for bootstrap values if the same alignment was used to reconstruct a phylogeny under alternate methods such as Maximum Likelihood. In my experience, while branches with bootstrap support of >60% in Maximum likelihood tend to have relatively stable position in a phylogeny as sequences are added or subtracted, you don't get a similar stability of branch position until a Bayesian posterior probability is over 90%. This is borne out here by the fact that the AK group and the base of the greater AK/Delta/AMPA/Kainate branch, which contain several posterior probabilities below 90%, are not stable between the two phylogenies. The Bayesian approach is a valid one, but here it seems unable to resolve one of the key groups that the authors propose. It isn't a valid approach to simply remove the ctenophore channels to get the desired result. Confidence in the AK group could be increased if it could be recovered by an alternate method of analysis such as Maximum Likelihood, and/or was recovered with high support in phylogenies based on amino acid sequences (point 1).

5) The methods of receptor identification and species selection are reasonable for identifying most of the glutamate receptors in a species or phylogenetic group, but they are by no means exhaustive. Receptors will be missed if automated gene predictions miss key exons or miss genes entirely. While this is not a major issue for the phylogenetic classification of receptors that have been found, it is a major problem for claims of receptor loss or absence in various phylogenetic groups. For example, the lack of NR2 subunits in *Nematostella* is used to claim a loss of NR2 in Cnidarians. While this might end up being true, the claim requires use of all data available from *Nematostella* (genome and transcriptomes in addition to gene predictions) and at least several of the other phylogenetically-diverse cnidarians for which data is available. If still nothing is found, then it may be safe to propose absence across this phylogenetic group. In this particular case, it isn't even clear if the absence of NR2 subunits would represent a loss, as the phylogeny could be interpreted to say that NR2 subunits simply evolved after the split between cnidarians and bilaterians. Throughout the paper, claims of loss in various groups should be reviewed to determine if data has been analyzed sufficiently to make the claim. If not, the claim in question could be removed or additional searches could be conducted to support the claim.

Reviewer #3

This paper presents a comprehensive phylogenetic analysis of metazoan glutamate receptor evolution, resulting in discovery of new families of glutamate receptor ion channels and GPCRs in phyla excluded from previous studies. Using amphioxus as a model system, mRNA expression in native tissue is established for some of these novel receptors via quantitative PCR, and using heterologous expression systems, cell surface expression and functional ion channel activity is established for one of the novel ion channels.

The work is of interest because it gives deeper insight into the evolution of this important class of neurotransmitter receptor than prior studies, which had largely focused on individual phyla, e.g. recent papers on ctenophores, molluscs or insects. Many of these studies had previously reported either receptors with novel properties (e.g. glycine activated homomeric ctenophore iGluRs), and/or the expansion of individual gene families in selected phyla, but did not do so in the context of a comprehensive analysis of metazoan phyla as presented here, and had failed to identify the AK and AKD1 families reported here.

As detailed below, there are issues in the current manuscript which in combination preclude publication until addressed. These include the need to provide a better presentation and interpretation of sequence alignment data; a more complete description of the sequences used for analysis; in addition, there are numerous errors which in places make the data presented very challenging to interpret. All of this can be addressed by careful rewriting.

1) It is unclear why the authors essentially exclude ctenophore iGluRs from their classification. In their revised scheme they identify eight iGluR families expressed in animals, but for a truly comprehensive study of metazoan iGluRs this needs revision to include at least two additional ctenophore subtypes, which are activated by glutamate and glycine respectively. Figure 1 should be replaced by the current Figure 1—figure supplement 4 using appropriate shading.

2) In multiple cases the sequence alignments are very difficult to interpret because they are presented without any shading to indicate identity/conservation and in addition lack coloring to indicate biochemically important residues (e.g. Cys residues known to form disulfide bonds in iGluRs of known structure etc). This should be addressed for ALL figures containing sequence alignments as commonly done by other groups. Notably, Figure 1—figure supplement 6 shows very poor conservation of the highly conserved M3 region highlighted in yellow for the four GluAk_Spu subunits, without any comment noting this.

3) Information should be given in a Table detailing the following for each of the novel AK, AKD1 and AKD2 predicted amino acid sequences: length (documenting unusual insertions or deletions separately for the ATD, LBD and TMD domains) and whether a predicted signal peptide is present and the predicted cut site. A good example can be found in Table S1 of Alberstein et al., 2015. For example, GluAk-beta_Nve appears to lack alpha helix D, a key structural element in the LBD.

4) I am concerned that for both of the novel iGluRs selected for heterologous expression, it was necessary to introduce a vertebrate signal peptide. Are these receptors actually expressed in native tissue? For GluAkd1-eta native expression is likely impossible because the signal peptide is a non consensus sequence that is much too short to form a signal peptide. For GluAkd1-alpha the sequence is much longer than normal, but the Sig3P server does predict cleavage.

5) Subsection “Phylogenetic identification of two unreported iGluR classes”. If it is most probable, as stated, that AKD2 is a long-branch attraction artifact does this warrant (i) inclusion of AKD2 as a separate family in Figure 1 and (ii) speculation that AKD2 may represent the ancestral genes of AMPA, kainate and Delta receptors? Revise text to make it clear that the earlier hypotheses that start this section are actually the least probable.

[Editors' note: further revisions were requested prior to acceptance, as described below.]

Thank you for resubmitting your work entitled "Evolution of Glutamate Receptors Reveals Three Unreported Classes and Divergent Phyla-Specific Adaptations" for further consideration at *eLife*. Your revised article has been evaluated by Richard Aldrich (Senior Editor), a Reviewing Editor, and two reviewers.

The manuscript has been improved but there are some remaining issues that need to be addressed before acceptance, as outlined below:

The authors comprehensively address the evolutionary diversity of ionotropic and metabotropic glutamate receptors at a level of detail that has not been previously done. They identify several new classes of receptor and provide numerous insights into the evolution of these gene families. In this revised manuscript, the authors have adequately addressed substantial concerns with the sequence search and phylogenetic methodologies. However, their discussion and interpretation of the phylogenies still have major issues that need to be addressed as outlined below:

1) The discussion and classification of what constitutes a new gene family here is somewhat arbitrary and confusing to follow, even though the data is quite clear. In the iGluRs, they have identified the Epsilon and Phi families, a cnidarian-specific clade of NMDA-like receptors and a cnidarian-specific sister group to the AMPA/Kainate/Delta/Phi group of bilaterians. These cnidarian-wide clades are mentioned at certain points in the manuscript, but the authors only headline the Epsilon and Phi families, even though Phi is more restricted in terms of phylogenetic spread than the cnidarian-specific groups. It might be better to say they have found one new type conserved across the metazoa (Epsilon), and 3 phylogenetically-restricted new types in cnidarians and deuterostomes. The authors should not arbitrarily state that their study has increased the number of iGluR classes from 6 to 8 when the data shows they have changed it from 6 to 10. Similarly, for the mGluRs their data shows that they have found: (1) that an mGluR type previously identified as insect-specific is widespread in bilaterians (they have not found a new family as stated, but interesting new information about an existing family), (2) three new cnidarian-specific classes, and (3) a class restricted to basal metazoan lineages including ctenophores, sponges, placozoans and perhaps cnidarians. Support for this latter class is just as strong as for any of the other classes, but it is puzzlingly not presented as a new class in the manuscript. The authors actually state that no classes of mGluRs span multiple phyla, but this ignored class does. The data seems to indicate that the number of mGluR types has moved from 3 to 8, not 3-4.

2) Throughout the manuscript, discussion of when the various types of receptors arose and what phyla/phylogenetic groups lost various receptor types is not interpreted correctly in light of recent advances in our understanding of the metazoan phylogeny. The current view is that ctenophores are the earliest-diverging metazoan phyla, followed by Sponge. Placozoans and Cnidarians group separately from ctenophores and sponges with the bilateria. Within this Placozoan/Cnidarian/Bilaterian group, the most accepted (though not universally accepted) order of divergence is placozoans > cnidarians > bilaterians. With this view of the animal phylogeny in mind: a) There is no evidence to support the claim that NMDA receptors were lost in ctenophores, sponges and placozoans. The simplest interpretation of the data is that NMDA receptors evolved after the divergence of these clades in an ancestor of cnidarians and bilaterians. This needs to be corrected. b) The Epsilon family, to which all ctenophore iGluRs belong, is likely to be the oldest and thus possibly the ancestral metazoan iGluR lineage. Epsilon must have first appeared in an ancestral metazoan prior to the divergence of ctenophores. It would thus be found in an ancestor of all extant metazoans, not just an ancestor of ctenophores/placozoans/cnidarians. Given the animal phylogeny above, that ancestor would also be the ancestor sponges and bilaterians.c) I can find no mention of sponges (porifera) in the results or discussion related to iGluRs. Are iGluRs not found in sponges? Or were they left out of the analysis? If the latter, that is a major oversight that needs to be corrected (and the only experiment I would see needing to be redone). If the former, the authors need to state that none were found, and then they could suggest that the Epsilon family might have been lost from sponges. There is no evidence presented that other iGluR classes were present prior to the divergence of sponges, so the absence of other classes in sponge would not represent gene loss. d) For the mGluRs, the ignored classes with sponge/ctenophore/placozoan/cnidarian sequences could be interpreted as the ancestral mGluR class, given that it is the only class present in the earliest diverging metazoan lineages.

3) The discussion of NMDA receptor evolution is not clear in all places. The data say that an NR2/NR3 ancestor evolved in the cnidarian/bilaterian ancestor. NR2 and NR3 then separated in a bilaterian ancestor after the divergence of bilaterians and cnidarians. In addition, cnidarians have a separate phyla-specific class of NMDA-like subunits. In Figure one, the cnidarian NR2/3 sequence and phyla-specific class are incorrectly lumped together as NMDA-like subunits. This needs to be corrected.

4) Alberstein (2015) reported that all ctenophore iGluRs contain a conserved disulfide bond in loop 1 of the LBD that is found also in NMDA receptors but not in AMPA receptors (their Figure 1, Figure 1—figure supplement 3 and Figure 1—figure supplement 4). This is an unusual structural feature, which combined with activation of some ctenophore iGluRs by glycine lead Alberstein to speculate that these were NMDA receptor precursors. Is this feature found in all of the Epsilon class? If not, is it conserved in sub branches of this clade?

---

## [Author Response]

Summary:This manuscript presents an extensive phylogenetic analysis of ionotropic and metabotropic glutamate receptors. The authors find two new families of iGluRs a and one family of mGluRs. Based on these new families, they also propose a new evolutionary tree for both classes of receptors. This work also provides functional data on one of the newly found families, AKD1 and show that they can work as excitatory glutamate receptors and are expressed in the nervous system of amphioxus, an early vertebrate.Essential revisions:Three reviewers have seen your manuscript. Although there is enthusiasm for the new findings reported, the reviewers found that there are major deficiencies with the analysis that led to the proposal of the new families of ionotropic glutamate receptors. In particular, the fact that the proposed AK1 family only shows up in one of the genealogies seems troubling and might be an indication that the methods used to get the phylogenies might not be optimal. The consensus among reviewers is that the phylogenetic analysis should be repeated using more robust methods (see the reviews), in order to discard the possibility that the AK1 group might be an artifact. If this is the case, it is possible that the AKD1 group and the new metabotropic receptor family will still be present, which is a significant finding. The proposed changes should be carried out in order for the manuscript to be reconsidered.1) If amino acid sequences were used for phylogeny reconstruction (after alignment of nucleotide sequences), then this needs to be clarified in the methods. However, if DNA sequences were used during phylogeny reconstruction, then both phylogenetic analyses presented in Figure 1 need to be repeated to make sure the proposed new classes of receptor are recovered.2) For the expression experiments of AKD1 receptors, authors need to clarify if maybe these receptors do no not express functionally, since they seem to lack signal peptides.3) The analysis done to claim loss of certain receptors in some phyla (e.g. Cnidarians) need to be redone and the interpretation needs to be in line with the results of the analysis, since as it stands, seems overreaching.

General approach taken to address comments on the phylogenies of ionotropic and metabotropic glutamate receptors:

In the first place we want to sincerely thank the reviewers for their constructive positive and comments on our work. For the past four months we have worked very hard to address all of them. We present a revised article that includes extended versions of our previous analysis and a new phylogenetic study. The main conclusions of the article remain essentially unchanged.

These are the major revisions performed:

1) Phylogenies are now presented using protein sequences (Comment 2.1).

2) All trees were calculated using two independent methods, a Bayesian and a maximum likelihood (ML) approach. Bayesian phylogenies are presented as main figures and ML trees are given as supplementary figures. We have only considered for discussion in the paper those features that are common to both analyses. (Comment 2.4)

3) Protein sequences from ctenophores have been included in the final trees (Comments 2.1, 2.2 and 3.1).

4) We have extended our sequence homology searches in order to identify sequences that might have been missed previously (see the ‘Materials and methods’ section in the revised version of the manuscript for a description of the new search strategy). This has resulted in the identification of 9 additional iGluRs and 6 extra mGluRs from the species we presented in the previous version of the manuscript. Two of the newly found iGluRs have allowed us to identify the Delta class in Lophotrochozoans. The rest were from the cnidarian *N. vectensis* and have been key to identify two cnidarian-specific groups of iGluRs. (Comment 2.5).

5) We have reviewed all claims regarding i/mGluR class loss in a particular phylum. This has been done by incorporating more species from that phylum in the analysis and by performing taxon-specific BLAST searches in the NCBI database. We now have at least two species in each phylum, with the exception of placozoans, for which only *T. adhaerens* is available. As a result, the new version of the iGluR phylogeny is based on the comparison of sequences from 25 species, as opposed to the 15 used previously (see Author response image 1). In a similar manner, for the mGluR phylogeny we have incorporated 11 additional species. (Author response image 2).

**Author response image 1. respfig1:** Species used in iGluR phylogeny. In bold new species used in the present version of the manuscript.

**Author response image 2. respfig2:** Species used in the mGluR phylogeny. In bold new species used in the present version of the mansucript.

6) The identification of missed sequences and the use of new species have resulted in an iGluR phylogeny with 233 sequences as opposed to the previous one that presented 171. Similarly, the mGluR phylogeny now includes 123 sequences, while previously it contained 103.

7) We have renamed the AKD1 and AK iGluR classes. AKD1 is now referred to as ‘Epsilon’ and AK as ‘Phi’. We consider that these names are more appropriate since the fourth iGluR class discovered was named after the fourth Greek letter (Delta). We thus though that unreported classes could receive the names of subsequent Greek letters. Gene and protein names from the Epsilon class start with Grie/GluE and those of the Phi class by Grif/GluF.

8) Finally, we have incorporated a new phylogenetic analysis to infer the phylogenetic history of AMPA and Kainate classes in protostomes (Figure 1—figure supplement 2 and Figure 1—figure supplement 3; see Author response image 3 for a list of species used in this phylogeny).

**Author response image 3. respfig3:** Species used in the phylogeny of AMPA and Kainate classes in protosotomes.

9) Taking into account the three phylogenies we present sequences from 45 different species. In this new version of the manuscript we include, as ‘Source Data’, the protein alignments used for each phylogeny. These are: Figure 1—source data 4, Figure 1—source data 5 and Figure 4—source data 3.

Principal changes to the interpretation of the phylogenetic evolution of iGluRs with respect to the previous version of the manuscript:

1) All subunits of ionotropic glutamate receptors from ctenophores belong to the Epsilon class (former AKD1 class).

2) The Phi class (former AK) now incorporates sequences from hemichordates but no longer includes sequences from the cnidarian *N. vectensis*. Phi proteins are only present in echinoderms, hemichordates and non-vertebrate chordates. For this reason, we hypothesize that this class was formed in an ancestor of deuterostomes. The addition of more cnidarian species to the phylogeny has revealed the existence of a cnidarian-specific group of non-NMDA iGluRs that contains the former AKs from *N. vectensis*.

3) In addition to this group there is a second cnidarian-specific group of NMDA-like iGluRs.

4) In the previous phylogeny a *N. vectensis* gene grouped with NMDA3 proteins (GRIN3_A/B_Nve). Yet in the present phylogeny this protein (current name GluN2-3_Nve) appears as the pre-bilaterian orthologue of both NMDA2 and NMDA3 classes. As we have not been able to identify members of the NMDA2 or NMDA3 classes in any of the cnidarian species investigated, the current phylogeny indicates that neither class NMDA2 (as proposed in the previous version of the manuscript) nor class NMDA3 would exist in pre-bilaterians.

5) Proteins from the former AKD2 class no longer cluster together. Therefore, we no longer refer to them as potentially constituting another class.

6) The existence of cnidarian- and placozoan-specific groups of non-NMDA iGluRs indicates that the duplication leading to the AMPA/Kainate/Phi/Delta branch occurred in pre-bilaterians.

7) Changes in the Delta class: we have been able to identify members of the Delta class in a few (three) lophotrochozoan species. Among ecdysozoans, priapulids might have retained members of this class as well. However, our phylogeny indicates that arthropods and nematodes, which account for the vast majority of ecdysozoan species, have lost this class.

8) The current version of *S. purpuratus* (sea urchin) genome does not include genes coding for members of any of the three NMDA classes. The inclusion of *Acanthaster planci* (starfish) in the revised phylogeny has allowed to identify members of classes NMDA1 and NMDA3 in the Echinodermata phylum. Class NMDA2 might have been lost in echinoderms, genomic data from other echinoderm species could help clarify this issue.

9) Author response image 4 and Author response image 5 summarize the distribution of iGluR classes among metazoan phyla in the previous version of the manuscript and inthe current version, respectively.

**Author response image 4. respfig4:** Previous distribution of iGluR classes among metazoan phyla.

**Author response image 5. respfig5:** Current distribution of iGluR classes among metazoan phyla. # We could only find 3 species with members of the Delta class in Lophotrochozoans. * Among Ecdysosoans members of the Delta class are only found in Priapulids. We have not found genes coding for subunits of Delta receptors in arthropod or nematode species.

Main changes in the interpretation of the phylogenetic evolution of metabotropic glutamate receptors with respect to the previous version of the manuscript:

1) We have identified three cnidarian-specific groups of mGluRs.

2) We now report Class IV proteins in echinoderms.

3) We have found Class I members in a few ecdysozoan species, including priapulids. Previously, we could only identify Class I mGluRs in lophotrochozoans amongst protostomes.

4) Author response image 6 summarizes the present distribution of mGluR classes among metazoan phyla.

**Author response image 6. respfig6:** Current distribution of mGluR classes among metazoan phyla.

Our detailed responses to all issues raised by the three reviewers follow below.

Reviewer #1The paper by Ramos-Vicente and collaborators presents bioinformatics and functional data pertaining the discovery of new classes of both ionotropic and metabotropic glutamate receptors. The authors have incorporated these new families into a revised evolutionary tree of glutamate receptors. These findings are exciting and expand our knowledge of the evolution of these important membrane proteins.1.1) In the Introduction the following phrase needs clarification: Thesespecies present a genomic structure, gene sequence and gene contentresembling that of their ancestors…. We do not know the genome of their ancestors and therefore this is an inference.

We have rewritten the Introduction to clarify this point. This fragment now reads as follows:

“We have favored the use of more slowly-evolving species for the construction of the phylogenetic trees. These species are particularly amenable to phylogenetics (21-23) as they arguably present lower rates of molecular evolution than other organisms.”

1.2) Structural models of two proteins from the new AKD1 class from amphioxus where constructed using known iGluR structures. Although this is at first glance reasonable, a less biased approach should be taken. Models should also be produced letting the software choose the appropriate structural template. Also, the MolProbity scores should be given as well as a discussion of the probability that the sequences adjust to the templates.

To address this point models for the full-length proteins from the Epsilon (formerly AKD1) class were generated again with RaptorX (http://raptorx.uchicago.edu/), which automatically selects the appropriate template for modeling. In our hands, RaptorX outperforms all other 3D prediction software when dealing with distant homologs, as indicated also by the results of the latest CASP contests (http://raptorx.uchicago.edu/about/). For modeling the well-conserved ligand-binding domain of GluE1 we used SWISS-MODEL (https://swissmodel.expasy.org/). Here, from a large number of equally appropriate templates, based on sequence similarity alone, we have chosen the atomic-resolution structure (1.26 Å) of the topologically equivalent rat GluA2 domain, which had been refined to an R_factor_ of 12.9% (R_free_: 15.7%).

In response to the reviewer’s request, we include two pairs of Supplementary Tables (Figure 1—figure supplement 8) containing all relevant MolProbity information for models of full-length GluE1 (formerly GluAkd1_alpha) and GluE7 (GluAkd1_eta), before and after refinement with ModRefiner (https://zhanglab.ccmb.med.umich.edu/ModRefiner/). Another Supplementary Table (Figure 1—figure supplement 9) presents the MolProbity results for the 3D model of the Gly-bound ligand-binding domain (LBD) from GluE1. As can be seen, MolProbity values compare well with those of structures solved at the resolution of the respective templates. The quality of the generated models is underscored by their low P-values (P-value is the likelihood of a predicted model of being worse than the best of a set of randomly-generated models for the studied protein (or domain). Thus, the smaller the P-value, the higher quality the model).

1.3) In the Discussion section, there is no correlation between the number of iGluR genes and the nervous system complexity of an organism. Is it possible that this might reflect the fact that in lower organisms glutamate receptors also have non-synaptic functions, as chemoreceptors?

We agree with the reviewer that some of these unreported receptors might perform functions other than behaving as neurotransmitter receptors; although most of them present good conservation in the residues involved in glycine or glutamate binding. Thus, to account for this possibility we have added the following text to the Discussion section:

“Whether all these proteins are expressed at the synapse and act as neurotransmitter receptors is an issue that will require further investigation. While, at least in amphioxus, all iGluRs are expressed in the central nervous system, their presence in other tissues, such as sensory organs, cannot be ruled out. Those receptors showing more divergent sequences, particularly in residues involved in ligand binding, might respond to molecules that are not neurotransmitters. For instance, they could behave as chemoreceptors in sensory organs, as it is the case of antennal receptors found in insects.

1.4) Please include more information on the iGluRs found on A. thaliana and their relation to the newly proposed phylogeny of iGluRs.

We have not identified the *Arabidopsis thaliana* iGluRs used in the phylogeny. These had been previously reported (please see Chiu et al., 2002 for a phylogenetic characterization of *A. thaliana* iGluRs). In the revised version of the manuscript we include this reference. We apologize for not having properly referenced these proteins in our previous version of the manuscript. Furthermore, ionotropic glutamate receptors from *A. thaliana* had been previously used as an outgroup for phylogenetic analysis of *C. elegans* iGluRs (please see Brockie et al., 2001, in our manuscript). It is for this reason that we choose to use them for the same purpose.

1.5) In the figures that plot mean values, it is not specified what measure of variability is being used (i.e. standard error, variance, etc.).

We have included this information in the legends to all figures presenting mean values (Figure 2, Figure 3 and Figure 4—figure supplement 2B).

Reviewer #2Conceptual IssuesIn the manuscript "Evolution of Glutamate Receptors Reveals Three Unreported Classes and Divergent Phyla-Specific Adaptations", the authors conduct a phylogenetic analysis of animal glutamate receptors (both ionotropic and metabotropic) and claim to have found 3 new phylogenetically-distinct classes (2 ionotropic and 1 metabotropic). They propose that the classification mechanism for glutamate receptors must be revised to account for these new gene families. Further, they provide data from structural modeling and functional expression to argue that ionotropic receptors in one of these classes, AKD1, are gated by glycine and not glutamate.While the manuscript addresses and interesting subject worthy of a high profile publication, there are issues with the phylogenetic analysis that must be addressed or clarified to understand how robustly the claims for new classes of ionotropic glutamate receptors are supported.2.1) There appears to be a major flaw in the methods for constructing the phylogeny, at least based on my interpretation of the Materials and methods section. The authors state that they aligned coding DNA sequences in MEGA6 with the "codon option" activated. The codon option will avoid introducing gaps into codons, but it does not correct for systematic biases that can get introduced into protein phylogenies that cover a wide range of divergent species with distinct codon usage biases. For this reason, it is generally accepted that such phylogenies need to be constructed with direct use of the amino acid sequences, as stated in the documentation for MEGA6. If amino acid sequences were used for phylogeny reconstruction (after alignment of nucleotide sequences), then this needs to be clarified in the methods. However, if DNA sequences were used during phylogeny reconstruction, then both phylogenetic analyses presented in Figure 1 need to be repeated to make sure the proposed new classes of receptor are recovered.

In the previous version of the manuscript we presented trees constructed with DNA sequences. This has been addressed and now we only present phylogenies constructed with protein sequences. The ‘Materials and methods’ section has been re-written accordingly in the revised version of the manuscript (please see subsection “Phylogenetic analyses”).

2.2) There are major issues with congruency between the phylogenies presented in Figure 1 with and without the ctenophore glutamate receptors, and this directly impacts the proposed new groups that the authors identify. Specifically, the AK and AKD2 groups in the main figure are split do not hold up in the analysis with the ctenophores included. While AKD2 is not a firmly-proposed new class, AK is, and this is concerning. It certainly suggests that proposing the AK class is an over-reach of the data.

We now include sequences from ctenophores in the final phylogenies of iGluRs and mGluRs. Furthermore, for each phylogenetic analysis we present trees constructed with a Bayesian and a maximum likelihood method (previously we only used the Bayesian approach). We have not found any relevant congruency issues between the two methods used in any of the three phylogenies presented (please see Figure 1 and Figure 1—figure supplement 1; Figure 1—figure supplement 2 and Figure 1—figure supplement 3; and Figure 4 and Figure 4—figure supplement 1).

As just mentioned, the revised iGluR phylogeny now includes sequences from the ctenophore *M. leidyi* (used previously). However we also include: (i) another member of the same phylum (*P. bachei*), (ii) three other cnidarians (*O. faveolata, A. digitifera* and *H. magnipapillata*), (iii) a second hemichordate (*P. flava*), (iv) a second echinoderm (*A. planci*), (v) a third cephalochordate (*B. floridae*), (vi) a member of the ecdysozoan lineage of priapulids (*P. caudatus*) and (vii) two other lophotrochozoans: *A. californica* and *C. gigas*.

The inclusion of ctenophores, as well as all these other species, in the phylogeny of iGluRs has resulted in an unexpected finding, namely, that all ctenophore iGluRs would in fact belong to the Epsilon class. In this revised version of the phylogeny we have also identified two cnidarian-specific groups of iGluRs.

Finally, we no longer present the AKD2 group as a potential new class.

2.3) The two phylogenies do not use the same names for the majority of the same genes. This needs to be corrected so that the position of a gene within the two phylogenies can be readily discerned. For instance, in the proposed AKD1 group, two genes in the phylogeny with the ctenophores simply contain XP numbers.

This issue has been addressed. Bayesian and maximum likelihood phylogenies present the same protein names in the revised version of the manuscript.

2.4) Posterior probabilities calculated in Bayesian phylogeny reconstructions are not equivalent to Bootstrap support as the authors suggest – this should be corrected. Furthermore, posterior probabilities tend to be much higher than one would get for bootstrap values if the same alignment was used to reconstruct a phylogeny under alternate methods such as Maximum Likelihood. In my experience, while branches with bootstrap support of >60% in Maximum likelihood tend to have relatively stable position in a phylogeny as sequences are added or subtracted, you don't get a similar stability of branch position until a Bayesian posterior probability is over 90%. This is borne out here by the fact that the AK group and the base of the greater AK/Delta/AMPA/Kainate branch, which contain several posterior probabilities below 90%, are not stable between the two phylogenies. The Bayesian approach is a valid one, but here it seems unable to resolve one of the key groups that the authors propose. It isn't a valid approach to simply remove the ctenophore channels to get the desired result. Confidence in the AK group could be increased if it could be recovered by an alternate method of analysis such as Maximum Likelihood, and/or was recovered with high support in phylogenies based on amino acid sequences (point 1).

In the previous version of our manuscript we inadvertently referred to posterior probabilities as bootstraps. This has now been corrected (Please see figure legends of phylogenetic trees).

For all phylogenetic analysis we now present two trees, one using a Bayesian approach and a second based on a Maximum likelihood method. In the two iGluR trees construed with these two different methods previously unreported classes are recovered and they have the same position in both trees. Including class AK, currently refer to as class Phi.

2.5) The methods of receptor identification and species selection are reasonable for identifying most of the glutamate receptors in a species or phylogenetic group, but they are by no means exhaustive. Receptors will be missed if automated gene predictions miss key exons or miss genes entirely. While this is not a major issue for the phylogenetic classification of receptors that have been found, it is a major problem for claims of receptor loss or absence in various phylogenetic groups. For example, the lack of NR2 subunits in Nematostella is used to claim a loss of NR2 in Cnidarians. While this might end up being true, the claim requires use of all data available from Nematostella (genome and transcriptomes in addition to gene predictions) and at least several of the other phylogenetically-diverse cnidarians for which data is available. If still nothing is found, then it may be safe to propose absence across this phylogenetic group. In this particular case, it isn't even clear if the absence of NR2 subunits would represent a loss, as the phylogeny could be interpreted to say that NR2 subunits simply evolved after the split between cnidarians and bilaterians. Throughout the paper, claims of loss in various groups should be reviewed to determine if data has been analyzed sufficiently to make the claim. If not, the claim in question could be removed or additional searches could be conducted to support the claim.

In order to perform a more exhaustive search we have undertaken the following approach: In addition to our previous search we have performed a reciprocal Blast search using TBLASTN and BLASTX. Protein sequences from mouse i/mGluRs were searched against nucleotide databases (genomic, as well as transcriptomic) using TBLASTN. The hits from this search were re-blasted using BlastX against the NCBI ‘non-redundant protein sequences’ database. Those query sequences that return i/mGluRs in the first position of the reciprocal Blast were kept for the phylogeny. Subsection “Identification of genes coding for members of glutamate receptor families in metazoan genomes” has been rewritten as follows:

“GluR sequences were identified using homology-based searches in a two-tier approach. Mouse ionotropic glutamate receptors were used as search queries (iGluRs: Gria1-4; Grik1-5; Grid1-2, Grin1, Grin2A-D and Grin-3A-B; mGluRs: mGluR1-8). In a first search GluR homologs were identified using the BLASTP tool with default parameters. Subject sequences with an E-value below 0.05 were selected as candidate homologs. These were re-blasted against the NCBI database of ‘non-redundant protein sequences’ using the same BLAST tool. If the first hit obtained in the reciprocal BLAST was a glutamate receptor the sequence was included in the phylogenetic analysis. In a second stage the same mouse sequences were used to perform TBLASTN searches against genomic and, when available, transcriptomic databases. Subject sequences not identified in the first tear and having an E-value below 0.05 were selected as candidate homologs. These were re-blasted using BLASTX against the NCBI ‘non-redundant protein sequences’ database. Finally, if the first hit of this search was a glutamate receptor the sequence was also included in the phylogenetic analysis.”

We have also included more cnidarian species in our analyses to account for possible bias caused by using only *Nematostella vectensis* (sea anemone). These are: *Acropora digitifera* (stony coral), *Hydra magnipapillata* (hydrozoa), *Orbicella faveolata* (stony coral), *Exaiptasia pallida* (sea anemone) and *Stylophora pistillata* (stony coral).

We have reviewed all claims regarding class loss in a particular phylum. This has been done by including more species from that phylum in the analysis and by performing taxon-specific Blast searches in the NCBI database. In the previous version we used 15 species for iGluR phylogenetic reconstruction and 18 for the mGluR tree. Now we use 25 and 29 species, respectively (please see Figure 1—source data 2 and Figure 4—source data 2, for a list of all species used in i/mGluR phylogenies).

Reviewer #3This paper presents a comprehensive phylogenetic analysis of metazoan glutamate receptor evolution, resulting in discovery of new families of glutamate receptor ion channels and GPCRs in phyla excluded from previous studies. Using amphioxus as a model system, mRNA expression in native tissue is established for some of these novel receptors via quantitative PCR, and using heterologous expression systems, cell surface expression and functional ion channel activity is established for one of the novel ion channels.The work is of interest because it gives deeper insight into the evolution of this important class of neurotransmitter receptor than prior studies, which had largely focused on individual phyla, e.g. recent papers on ctenophores, molluscs or insects. Many of these studies had previously reported either receptors with novel properties (e.g. glycine activated homomeric ctenophore iGluRs), and/or the expansion of individual gene families in selected phyla, but did not do so in the context of a comprehensive analysis of metazoan phyla as presented here, and had failed to identify the AK and AKD1 families reported here.As detailed below, there are issues in the current manuscript which in combination preclude publication until addressed. These include the need to provide a better presentation and interpretation of sequence alignment data; a more complete description of the sequences used for analysis; in addition, there are numerous errors which in places make the data presented very challenging to interpret. All of this can be addressed by careful rewriting.3.1) It is unclear why the authors essentially exclude ctenophore iGluRs from their classification. In their revised scheme they identify eight iGluR families expressed in animals, but for a truly comprehensive study of metazoan iGluRs this needs revision to include at least two additional ctenophore subtypes, which are activated by glutamate and glycine respectively. Figure 1 should be replaced by the current Figure 1—figure supplement 4 using appropriate shading.

We now include two ctenophora species in the final phylogenies of iGluRs and mGluRs. To better account for this phylum we include sequences from *Pleurobrachia bachei*, as before we only used sequences from *M. leidyi*. Three of the previously reported proteins from P. bachei (Alberstein et al., 2015) did not present a sufficiently conserved SYTANLAAF motif in our alignments (please see Materials and methods section). For this reason, they were not included in the final tree, these are: PbiGluR5, PbiGluR11 and PbiGluR12. As we now incorporate most iGluRs from these two ctenophore species into the phylogeny we do include members of two subtypes of ctenophore iGluRs (activated by glutamate and glycine) as requested by the reviewer.

3.2) In multiple cases the sequence alignments are very difficult to interpret because they are presented without any shading to indicate identity/conservation and in addition lack coloring to indicate biochemically important residues (e.g. Cys residues known to form disulfide bonds in iGluRs of known structure etc). This should be addressed for ALL figures containing sequence alignments as commonly done by other groups. Notably, Figure 1—figure supplement 6 shows very poor conservation of the highly conserved M3 region highlighted in yellow for the four GluAk_Spu subunits, without any comment noting this.

We thank the reviewer for this comment. We have amended all figures with sequence alignments following the reviewer indications.

Regarding the conservation of the SYTANLAAF motif in GluAk_Spu (currently GluF_Spu), we want a first mention that in order to be more systematic we have now established as a rule that only sequences with at least 4 out of 9 residues from this motif conserved are included in phylogenetic analysis. This is mentioned in the corresponding ‘Materials and methods’ section. Because of this criterion we had to remove one of the four *S. purpuratus* AKs (GluAk_delta_Spu) from the final phylogeny, as it only has 3/9 residues conserved in this motif. Nevertheless, as the reviewer mentions, Phi sequences (former AKs) from echinoderms present weak conservation in this motif, although most residue changes are quite conservative. Indeed, the sequences from hemichordates that we now present as belonging to class Phi (GluF_Sko and GluF_Pfl), also have low levels of conservation in this motif. We have added a few sentences in the Results sections to refer to this observation. We have not gone into a detailed discussion about these changes as we believe that this is out of the scope of the present manuscript. Modified text:

“The ‘SYTANLAAF’ motif, which is essential for channel gating (7), is also well conserved in most sequences. Nevertheless, some members of the Phi class and the cnidarian NMDA-like group present lower levels of conservation in this sequence. Whether these changes have a functional impact is something that will require further investigation.”

3.3) Information should be given in a Table detailing the following for each of the novel AK, AKD1 and AKD2 predicted amino acid sequences: length (documenting unusual insertions or deletions separately for the ATD, LBD and TMD domains) and whether a predicted signal peptide is present and the predicted cut site. A good example can be found in Table S1 of Alberstein et al., 2015. For example, GluAk-beta_Nve appears to lack alpha helix D, a key structural element in the LBD.

We now include Tables detailing all this information, as suggested by the reviewer (please see Figure 1—source data 1, Figure 4—source data 1). When there is a difference in domain length between mammalian proteins and non-vertebrate i/mGluRs this is, in most cases, because non-vertebrate proteins are shorter. We think that these ‘potential deletions’ are in fact due to poor genome sequence data, as they are usually very large, especially in the ATD domain. For instance, the sequence of GluAKbeta_Nve (now GluCni2_Nve), that the reviewer refers to, only presents 140 LBD residues (while mammalian ones present >250).

3.4) I am concerned that for both of the novel iGluRs selected for heterologous expression, it was necessary to introduce a vertebrate signal peptide. Are these receptors actually expressed in native tissue? For GluAkd1-eta native expression is likely impossible because the signal peptide is a non consensus sequence that is much too short to form a signal peptide. For GluAkd1-alpha the sequence is much longer than normal, but the Sig3P server does predict cleavage.

We provide experimental evidence for the expression of the genes Grie1 and Grie7 in the cephalochordate *B. lanceolatum* (Figure 2A); these genes code for GluE1 (formerly GluAkd1_α) and GluE7 (GluAkd1_η), respectively. Importantly, we show that these two genes present an enriched expression in the nervous cord of amphioxus, indicating that their main biological function is performed in this tissue. Furthermore, the expression level of Grie1 is similar to that observed for genes coding for subunits of AMPA, Kainate or NMDA receptors from amphioxus (Figure 2A).

Secondly, as indicated by the reviewer, a signal peptide (SP) can be identified in GluE1, although this is longer than canonical SPs found in vertebrates. Nevertheless, the existence of long SPs, of over 50 residues, is well accepted (Nielsen, 2017) and therefore SignalP includes an option to predict them. A simple explanation for the lack of a SP in GluE7, as predicted by SignalP4.1, might be that the 5’ end of this gene has not been completely sequenced, missing its SP.

Nevertheless, we have also considered the possibility that SignalP4.1 might be less efficient at identifying SPs from non-vertebrates. We have consulted Dr. Henrik Nielsen, the key scientist responsible for developing this software, who has communicated us that invertebrate sequences are less represented in the training sets used to develop SignalP. He has also explained us that SignalP 4.1 is less efficient at identifying long SPs than SPs with canonical lengths.

These observations prompted us to compare SignalP performance with iGluR sequences from the vertebrate and invertebrates species used in this article. To avoid bias due to the different completeness of vertebrate and non-vertebrate genomes, we have only considered proteins with at least 800 residues that start with a methionine, as these most likely have a well preserved N-terminus. Our results are summarized in Author response image 7:

**Author response image 7. respfig7:** 

This analysis suggests that SignalP4.1 is not as efficient at identifying SPs in non-vertebrate iGluRs as it is for vertebrate ones. The increase in SPs predicted when long SPs are allowed suggests that invertebrate species would have a higher proportion of iGluRs with long SPs. Nevertheless, differences in genomic data quality between vertebrate and invertebrate species might explain part of the differences observed.

In our mRNA expression analysis (Figure 2A) all iGluRs from amphioxus were found expressed in the nervous cord, yet SignalP4.1 fails to identify a SP in approximately half of them. Importantly, SignalP4.1 not only fails to identify SPs in members of the new classes reported in this article, it also misses SPs in proteins from known classes, such as in Kainate or NMDA2 (see Author response image 8).

**Author response image 8. respfig8:** 

Altogether these observations suggest that SignalP4.1 is less efficient at predicting SPs in invertebrate species, which in turn might present more iGluRs with long SPs. This is in accordance with the lower number of sequences from non-vertebrate species available for the training sets used to develop SignalP, as reported by Dr, Nielsen.

Therefore, the fact that SignalP4.1 does not predict a SP in an invertebrate iGluR should not be taken as a very strong indication for the absence of a functional SP. Unfortunately, to the best of our knowledge there are no comprehensive studies on the evolution/conservation of SPs along the metazoan kingdom, and the evolution of the machinery involved in SP processing has neither been systematically investigated. Thus, we do not know if all SPs from non-vertebrates would be recognized by mammalian cells (in our particular case human HEK293 cells). It is plausible that the mammalian machinery involved in SP identification and protein trafficking to the plasma membrane is not able to properly recognize SPs from cephalochordate proteins, which diverged from the main metazoan lineage over 650 million years ago (TimeTree: http://www.timetree.org).

Finally, we also want to indicate that GluE1 and GluE7 both expressed well in HEK293 cells without the SP from rat GluA2, although they were not efficiently trafficked to the plasma membrane (please see Author response image 9, and Figure 2—figure supplement 1C,D for immunoblots of whole HEK293 cell lysates 48 hours post-transfection of wild-type GluE1 and GluE7). Note that protein degradation is not observed, which generally is a good indicator of protein stability. In our opinion, the fact that these two proteins, despite their considerable degree of divergence from mammalian homologues (the average protein identity between members of the Epsilon class from amphioxus and mouse iGluRs is around 25%) can be efficiently expressed in HEK293 cells suggests that they are functional, that is, that they are expressed in native tissue. In this regard, we wish to note that non-expressed genes rapidly accumulate deleterious mutations becoming pseudogenes that are usually not identified by standard genomics projects.

**Author response image 9. respfig9:** Expression of GluE1 and GluE7 in HEK293 48 hours after transfection.

We therefore consider that the fact that wild-type GluE1 and GluE7 could not be trafficked to the plasma membrane of HEK293 cells nor their lack of canonical SPs (as predicted by SignalP4.1) warrant sufficient evidence to discard their expression in native tissue. On the contrary, we think that the following evidence strongly supports their functional expression in native tissue: (i) we do observe expression of both genes at the mRNA level, (ii) GluE1 has a SP, although a long one, (iii) GluE1 presents expression levels similar to those of GluAs, GluKs or GluNs from amphioxus, (iv) both proteins express well in HEK293 cells and, finally, (v) GluE1 forms functional ligand-gated ion channels.

3.5) Subsection “Phylogenetic identification of two unreported iGluR classes”. If it is most probable, as stated, that AKD2 is a long-branch attraction artifact does this warrant (i) inclusion of AKD2 as a separate family in Figure 1 and (ii) speculation that AKD2 may represent the ancestral genes of AMPA, kainate and Delta receptors? Revise text to make it clear that the earlier hypotheses that start this section are actually the least probable.

In the revised iGluR phylogeny we no longer consider the AKD2 class.

[Editors' note: further revisions were requested prior to acceptance, as described below.]

The manuscript has been improved but there are some remaining issues that need to be addressed before acceptance, as outlined below:The authors comprehensively address the evolutionary diversity of ionotropic and metabotropic glutamate receptors at a level of detail that has not been previously done. They identify several new classes of receptor and provide numerous insights into the evolution of these gene families. In this revised manuscript, the authors have adequately addressed substantial concerns with the sequence search and phylogenetic methodologies. However, their discussion and interpretation of the phylogenies still have major issues that need to be addressed as outlined below:1) The discussion and classification of what constitutes a new gene family here is somewhat arbitrary and confusing to follow, even though the data is quite clear. In the iGluRs, they have identified the Epsilon and Phi families, a cnidarian-specific clade of NMDA-like receptors and a cnidarian-specific sister group to the AMPA/Kainate/Delta/Phi group of bilaterians. These cnidarian-wide clades are mentioned at certain points in the manuscript, but the authors only headline the Epsilon and Phi families, even though Phi is more restricted in terms of phylogenetic spread than the cnidarian-specific groups. It might be better to say they have found one new type conserved across the metazoa (Epsilon), and 3 phylogenetically-restricted new types in cnidarians and deuterostomes. The authors should not arbitrarily state that their study has increased the number of iGluR classes from 6 to 8 when the data shows they have changed it from 6 to 10. Similarly, for the mGluRs their data shows that they have found: (1) that an mGluR type previously identified as insect-specific is widespread in bilaterians (they have not found a new family as stated, but interesting new information about an existing family), (2) three new cnidarian-specific classes, and (3) a class restricted to basal metazoan lineages including ctenophores, sponges, placozoans and perhaps cnidarians. Support for this latter class is just as strong as for any of the other classes, but it is puzzlingly not presented as a new class in the manuscript. The authors actually state that no classes of mGluRs span multiple phyla, but this ignored class does. The data seems to indicate that the number of mGluR types has moved from 3 to 8, not 3-4.

We regret that our classification was confusing to follow. In order to resolve this problem we now present a different classification system that we think is clearer as well as more accurate. As requested by the reviewers we have included in our lists of ‘Classes’ phylogenetic groups found in only one phylum, which in the previous version of the manuscript were classified as ‘Groups’. This modification together with the inclusion of porifers in the phylogenies has resulted in the identification of 10 iGluR classes and 8 mGluR classes.

In iGluRs the classification system presents 3 levels: family, subfamily and class. In mGluRs only 2, family and class. Although there is not a total agreement in the field of neurosciences, we believe that the term ‘family’ is generally used to refer to all iGluRs or all mGluRs. Thus, we reserve the use of ‘family’ for this porpoise; there are therefore only 2 families of GluRs (iGluRs and mGluRs).

We now give more strength to the idea that metazoan iGluRs are organized into 4 monophyletic groups. Three of these groups were already present with the same topology in our previous version of this manuscript, although we did not discuss them as such. The incorporation of porifers has resulted in the addition of a fourth main group to the previous three. We have given the name of ‘subfamily’ to each of these groups, to distinguish them from ‘classes’. Classes are contained within subfamilies. Due to the topology of both iGluR trees (Bayes and ML) we propose that these 4 subfamilies arose from 3 gene duplication events that occurred prior to the radiation of all present-day metazoan phyla (please see Figure 1).

Two of these subfamilies include the well-known vertebrate classes. The first one, subfamily ‘NMDA’, includes classes NMDA1, NMDA2, NMDA3, NMDA2/3 and NMDA-Cnidaria. The other subfamily, which we have termed ‘AKDF’, has 5 classes: AMPA, Kainate, Delta, Phi and AKDF-Oca. The later exclusively contains sequences from one sponge (*O. carmela*), but it is present in the Bayes and ML phylogenies with very good statistical support. The same is true for class NMDA2/3, for which we have only identified 1 sequence from *N. vectensis*. Following reviewers’ advice, we now refer to these groups as ‘classes’.

All these classes (but AKDF-Oca) were already present in the previous version of our manuscript containing the same proteins and presenting the same topology. Beyond AKDF-Oca we have found several proteins from porifer, placozoan and cnidarian species that clearly belong to the AKDF subfamily. There are two important considerations about these proteins. In the first place one of them (from *T. adhaerens*, in green) is in the branch of Deltas in the Bayes tree. Yet the ML tree does not place this sequence in this class. Therefore, we do not consider it as being a Delta member. In the second place, the branches involving these sequences present different topologies in the Bayes and ML trees. For this reason, we have left them unclassified.

The former Epsilon class is now referred to as Epsilon subfamily, as it is at the same phylogenetic level as the other subfamilies. Finally, as mentioned earlier, the addition of sequences from sponges has allowed identifying a fourth subfamily, being located at the very base of the tree, that we have termed ‘Lambda’.

Finally, regarding the phylogeny of mGluRs, the addition of new poriferan species has introduced some changes regarding the organization of non-bilaterian classes. Please see below.

2) Throughout the manuscript, discussion of when the various types of receptors arose and what phyla/phylogenetic groups lost various receptor types is not interpreted correctly in light of recent advances in our understanding of the metazoan phylogeny. The current view is that Ctenophores are the earliest-diverging metazoan phyla, followed by Sponge. Placozoans and Cnidarians group separately from ctenophores and sponges with the bilateria. Within this Placozoan/Cnidarian/Bilaterian group, the most accepted (though not universally accepted) order of divergence is placozoans > cnidarians > bilaterians.

We are aware of the current debate in the field about which metazoan phylum is the earliest-diverging one. Several important publications (Moroz et al., 2014; Ryan et al., 2013 or Dunn et al., 2008, among others) advocate for ctenophores as the earliest metazoan phylum; yet, other authors support a Porifera-First model (please see, for instance, Simon et al., 2017 and Feuda et al., 2017). We believe that on the matter of which is the earliest-diverging metazoan phylum, the jury is still out, and that unanimous consensus has not yet been reached. Nevertheless, in order to account for the recommendations of the reviewers, we have interpreted our phylogenies considering a ctenophores-first model.

With this view of the animal phylogeny in mind: a) There is no evidence to support the claim that NMDA receptors were lost in ctenophores, sponges and placozoans. The simplest interpretation of the data is that NMDA receptors evolved after the divergence of these clades in an ancestor of cnidarians and bilaterians. This needs to be corrected.

We disagree with this remark. We believe that since the branch of the NMDA subfamily appears at the same level that the branch with ctenophore iGluRs (See Figure 1), NMDA protein/s must have appeared before the divergence of Ctenophores and lost in this phylum.

b) The Epsilon family, to which all ctenophore iGluRs belong, is likely to be the oldest and thus possibly the ancestral metazoan iGluR lineage. Epsilon must have first appeared in an ancestral metazoan prior to the divergence of ctenophores. It would thus be found in an ancestor of all extant metazoans, not just an ancestor of ctenophores/placozoans/cnidarians. Given the animal phylogeny above, that ancestor would also be the ancestor sponges and bilaterians.

While we realize that the possibility presented by the reviewers is very parsimonious and would agree well with the ctenophores-first model of animal evolution, the present iGluR phylogeny indicates that there were 4 subfamilies of iGluRs in the last common ancestor to present-day metazoans: Epsilon, NMDA, AKDF and Lambda.

c) I can find no mention of sponges (porifera) in the results or discussion related to iGluRs. Are iGluRs not found in sponges? Or were they left out of the analysis? If the latter, that is a major oversight that needs to be corrected (and the only experiment I would see needing to be redone). If the former, the authors need to state that none were found, and then they could suggest that the Epsilon family might have been lost from sponges. There is no evidence presented that other iGluR classes were present prior to the divergence of sponges, so the absence of other classes in sponge would not represent gene loss.

Thanks to the identification of the Compagen Database (http://www.compagen.org), which we previously were not aware of, we could incorporate porifers into the iGluR phylogeny. We have also included poriferan species from Compagen into the mGluRs phylogeny.

d) For the mGluRs, the ignored classes with sponge/ctenophore/placozoan/cnidarian sequences could be interpreted as the ancestral mGluR class, given that it is the only class present in the earliest diverging metazoan lineages.

We agree that in our previous manuscript we did not discuss in detail the large branch close to the base of the tree including sponge, ctenophore, placozoan and cnidarian sequences. Although, as stated by the reviewers, this branch was consistently found in the Bayes and ML trees with good statistical support. The reason why we did not discuss it in detail is that we were puzzled by the presence of a more basal branch that only included cnidarian sequences (Cni3). We found difficult to integrate these two groups into a parsimonious explanation of the early evolution of mGluRs.

In any case, the incorporation of new porifers in the mGluR phylogeny has changed this part of the tree. Now the former Cni3 group has merged with another former cnidarian-specific group (Cni2) and is located next to the former Cni1 group. These two branches of cnidarian mGluRs are clearly orthologue to classes I-IV in the present phylogeny. Similarly, we found clear orthologues of these 4 classes in placozoans and sponges. Altogether these non-bilaterian orthologues of classes I-IV are organized in 4 branches, with identical topology in bates and ML trees (Figure 6 and Figure 6—figure supplement 1). Following reviewers’ advice, we now refer to these branches as classes. Thus, in total, we report 8 classes of mGluRs.

3) The discussion of NMDA receptor evolution is not clear in all places. The data say that an NR2/NR3 ancestor evolved in the cnidarian/bilaterian ancestor. NR2 and NR3 then separated in a bilaterian ancestor after the divergence of bilaterians and cnidarians. In addition, cnidarians have a separate phyla-specific class of NMDA-like subunits. In Figure one, the cnidarian NR2/3 sequence and phyla-specific class are incorrectly lumped together as NMDA-like subunits. This needs to be corrected.

As previously mention we have re-formulated the overall classification of iGluRs. We hope that the explanation regarding NMDA classes evolution is now clear. We have also amended the error in Figure 1.

4) Alberstein (2015) reported that all ctenophore iGluRs contain a conserved disulfide bond in loop 1 of the LBD that is found also in NMDA receptors but not in AMPA receptors (their Figure 1, Figure 1—figure supplement 3 and Figure 1—figure supplement 4). This is an unusual structural feature, which combined with activation of some ctenophore iGluRs by glycine lead Alberstein to speculate that these were NMDA receptor precursors. Is this feature found in all of the Epsilon class? If not, is it conserved in sub branches of this clade?

This is certainly a relevant point as it could have important evolutionary implications. It is something we looked into but did not include in the manuscript, as these two cysteines are clearly absent from all other Epsilon proteins. Furthermore, these two residues are also absent from all non-bilaterian AKDF proteins and most Lambda members. In our alignment (Figure 1—source data 3) the second (most C-term) cysteine forming this bond is very well aligned in all NMDAs and Epsilons from ctenophores. The only sequence among unreported groups presenting a good alignment of this cysteine is GluL1_Oca (a Lambda member from *O. Carmela*). This sequence also presents a cysteine 9 residues upstream, which could participate in a S-S bond. Nevertheless, experimental data will probably be needed to confirm if this bond is formed.

Yet, as this is an interesting point, we now refer to this sequential difference among Epsilon proteins in the Results section. We have added the following sentences:

“We have also identified a sequence difference among Epsilon proteins. Ctenophore iGluRs have two cysteines that form a disulfide bond at loop 1 of the ligand binding domain (26), which are also present in NMDA proteins. Nevertheless, this element is absent from the remaining members of the Epsilon subfamily.”